# ENHANCING ONE-SHOT FEDERATED LEARNING THROUGH DATA AND ENSEMBLE CO-BOOSTING

**Rong Dai**[1,2], **Yonggang Zhang**[2], **Ang Li**[3], **Tongliang Liu**[4], **Xun Yang**[1,*], **Bo Han**[2]
[1]University of Science and Technology of China, [2]TMLR Group, Hong Kong Baptist University
[3]ECE Department, University of Maryland College Park, [4]Sydney AI Centre, The University of Sydney
`rongdai@mail.ustc.edu.cn` `{csygzhang, bhanml}@comp.hkbu.edu.hk`
`angliece@umd.edu` `tongliang.liu@sydney.edu.au` `xyang21@ustc.edu.cn`

## ABSTRACT

One-shot Federated Learning (OFL) has become a promising learning paradigm, enabling the training of a global server model via a single communication round. In OFL, the server model is aggregated by distilling knowledge from all client models (the ensemble), which are also responsible for synthesizing samples for distillation. In this regard, advanced works show that the performance of the server model is intrinsically related to the quality of the synthesized data and the ensemble model. To promote OFL, we introduce a novel framework, Co-Boosting, in which synthesized data and the ensemble model mutually enhance each other progressively. Specifically, Co-Boosting leverages the current ensemble model to synthesize higher-quality samples in an adversarial manner. These hard samples are then employed to promote the quality of the ensemble model by adjusting the ensembling weights for each client model. Consequently, Co-Boosting periodically achieves high-quality data and ensemble models. Extensive experiments demonstrate that Co-Boosting can substantially outperform existing baselines under various settings. Moreover, Co-Boosting eliminates the need for adjustments to the client's local training, requires no additional data or model transmission, and allows client models to have heterogeneous architectures.

## 1 INTRODUCTION

Federated learning (FL) (McMahan et al., 2017) has emerged as a prominent distributed machine learning framework to train a global server model via collaboration among users without sharing their dataset. Though the multi-round parameter-server communication paradigm offers the benefit of effectively exchanging information among clients and the central server, it might not be feasible in the real world. This paradigm brings forth significant challenges: 1) heavy communication burden and the risk of connection drop errors between clients and the server (Li et al., 2020a; Kairouz et al., 2021; Dai et al., 2022), and 2) potential risk for man-in-the-middle attacks (Wang et al., 2021) and various other privacy or security concerns (Mothukuri et al., 2021; Yin et al., 2021).

One-shot FL (OFL) (Guha et al., 2019) has emerged as a solution to these issues by restricting communication rounds to a single iteration, thereby mitigating errors arising from multi-round communication and concurrently diminishing the vulnerability to malicious interception. Furthermore, OFL is more practical, particularly within contemporary model market scenarios (Vartak et al., 2016) where clients predominantly offer pre-trained models. In OFL, the server model is aggregated by distilling knowledge from all client models, commonly using the ensemble, while the ensemble is also responsible for synthesizing data samples for knowledge distillation. Consequently, as illustrated in Guha et al. (2019) and Zhang et al. (2022a), the server model's performance is intricately linked to both the quality of synthesized data and the ensemble. Thus, the primary challenge in improving performance lies in the process of improving the data and the ensemble.

Existing approaches tend to tackle this challenge by exclusively concentrating on either enhancing the quality of the ensemble or improving the quality of synthetic data. For instance, to bolster ensemble, prior works including Dennis et al. (2021), Heinbaugh et al. (2023), and Diao et al. (2023)

---

*Corresponding author. Work done during Rong's visit to TMLR Group at HKBU.

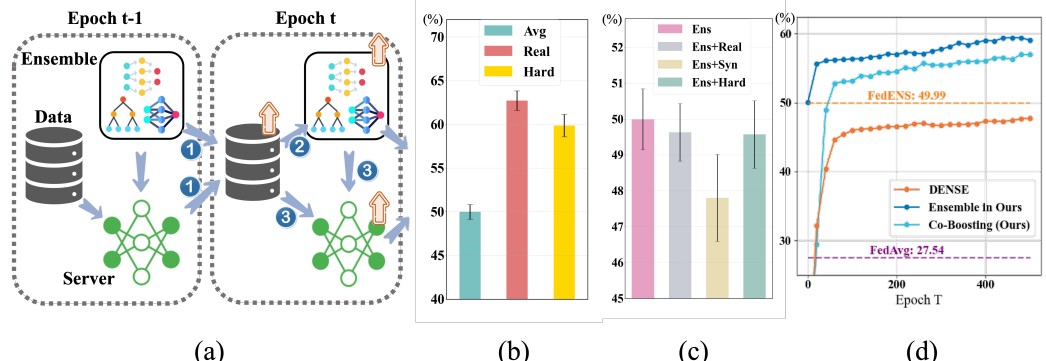

(a)        (b)        (c)        (d)

Figure 1: Co-Boosting Framework and Experimental Comparison. (a) illustrates the core concept of our approach. In each epoch, high-quality samples are first generated based on last epoch's ensemble and server, which are then used to adjust client weights giving a better ensemble. Based on the enriched data and refined ensemble, server model is updated by distilling knowledge from them. (b) shows test accuracy of ensemble with **averaged** weights, learned weights on **real** data, and learned weights on **hard** samples. (c) shows test accuracy of server obtained through distillation on **real**, **synthetic**, and **hard** samples with same ensemble. (d) presents an overall comparison. DENSE (Zhang et al., 2022a) signifies the current state-of-the-art, FedENS denotes the averaged ensemble. Experiments in (b)(c)(d) are all done on CIFAR-10 with a 10-client $Dir(0.1)$-parted setting.

modify the local training phase and require additional transmissions. In terms of improving synthetic data, Li et al. (2021) utilizes auxiliary public datasets, Zhou et al. (2020) proposes transmitting distilled datasets to the server, Yang et al. (2023) proposes to use auxiliary diffusion model and Zhang et al. (2022a) employs data-free data generation methods to synthesize data directly from averaged ensemble models. While the distilled server may improve through the above methods, it is noteworthy that these approaches typically follow a sequential process, which means the enhancement of data or the ensemble is a prerequisite step before the server model can benefit, omitting the crucial relationship between them. What's more, in contemporary model market scenarios where only well-pre-trained models with diverse architectural possibilities are accessible, any modifications to local training or additional data or model transmissions are discouraged and often disallowed.

To address these challenges, we propose Co-Boosting, a novel one-shot federated learning algorithm as in Fig. 1(a), in which the synthesized data and the ensemble model mutually boost each other progressively. More specifically, in each training epoch, higher-quality hard samples are generated based on the previous epoch's ensemble and server model. Based on these hard samples, the aggregation weight for each client model is adjusted, forming a better ensemble. Subsequently, the server model is updated by distilling knowledge from both the enriched data and the refined ensemble. As a result, with the continuous enhancement of both data and the ensemble, the final server model naturally improves. As depicted in Fig. 1(b), (c), and (d), with a better weighted ensemble model and better-quality hard samples, Co-boosting naturally achieves state-of-art performance.

Thorough experiments on multiple benchmark datasets demonstrate the superiority of the proposed Co-Boosting. What's more, due to its inherent nature, our proposed Co-Boosting is more practical to today's model market scenarios. In summary, our main contributions can be summarized as follows: 1) We demonstrate that it is possible to simultaneously improve the quality of the synthesized data and the ensemble, which are two key elements in OFL. This discovery could spur progress in OFL methods, highlighting the need to optimize their interaction. 2) Within an adversarial paradigm, we introduce Co-Boosting, a novel one-shot federated learning method. Periodically, in Co-Boosting, hard samples are generated from the current ensemble, which, in turn, are used to reweight clients, forming an improved ensemble. This mutual enhancement of synthetic data quality and the ensemble collectively contributes to the natural emergence of a high-performing distilled server model. 3) Our proposed method Co-Boosting, is highly practical to the contemporary model market scenarios as it eliminates the necessity for client-side training adjustments, entails no extra data or model transmissions, and accommodates diverse client model architectures. 4) Extensive experiments confirm the effectiveness of Co-Boosting, consistently outperforming other baselines thanks to the improved quality of both the synthetic data and ensemble.

## 2 Related Works

### 2.1 One-shot Federated learning

Guha et al. (2019) originally proposes OFL which collects local models as an ensemble for the final prediction and further proposes to use knowledge distillation (KD) on such ensemble with public data. This paradigm, which is followed by most works, inherently relates the performance of the server model to the data and ensemble used in the KD stage. Li et al. (2021) proposes to improve the ensemble on the public data. Instead of using public data, Zhou et al. (2020) proposes to transmit the distilled local dataset for the server, Yang et al. (2023) proposes to use auxiliary pre-trained diffusion model, while Zhang et al. (2022a) generates fake data scouring from the direct ensemble. Regarding the improvement of the ensemble, Dennis et al. (2021) utilizes a cluster-based method and requires uploading the cluster means. Diao et al. (2023) and Heinbaugh et al. (2023) modify the local training phase of each client by introducing placeholders, or conditional variation auto-encoders. However, none of the aforementioned methods simultaneously address improvements in both data and the ensemble. Moreover, few works can be practically applied, especially in contemporary model-market scenarios (Vartak et al., 2016) where only well-pretrained models are provided to the server. This situation implies constraints such as no alterations to the client's local training, no additional transmissions, and the possibility of client model heterogeneity.

### 2.2 Knowledge Distillation

Knowledge distillation (KD) (Hinton et al., 2015) is proposed to transfer knowledge from one or more networks (teacher) to another (student). Taking the same spirit, KD in federated learning focuses on transferring knowledge from multiple local clients to the global server model. Lin et al. (2020) initially introduced to utilize KD at the server side based on an unlabeled auxiliary dataset. In an effort to reduce reliance on proxy datasets, generators that are locally updated and globally aggregated are used in Zhu et al. (2021) and Zhang et al. (2022b) to synthesize distillation samples. Wang et al. (2023) further enhances the basic ensemble distillation by using weighted averaging based on locally trained discriminators. However, in the context of OFL, conducting multiple rounds of training or transmitting generators and discriminators is not practical. Additionally, the need for an additional local client component violates the constraints in modern model-market OFL settings. More seriously, the generator trained locally has direct access to the training samples, potentially leading to privacy leakage through its ability to remember all the training data (Liu et al., 2019). On the other hand, the generator in OFL is trained without access to even one single raw data.

## 3 Methodology

In this section, we first introduce the general process of one-shot federated learning (OFL). Then we detail the proposed method, Co-Boosting, in how we generate high-quality data, high-quality ensemble, and how to link and make them boost each other as illustrated in Fig. 1(a).

### 3.1 One-Shot Federated Learning

Suppose we have a set of clients $\mathbb{C}$, with $n = |\mathbb{C}|$ clients in total. Each client $c_k \in \mathbb{C}$ has a local private dataset $\mathbb{D}^k = \{(\mathbf{x}_i, y_i)\}_{i=1}^{n_k}$, where $n_k = |\mathbb{D}^k|$ is the number of local data samples $\mathbf{x}_i$ with the corresponding label $y_i$. OFL's goal is to train a good machine learning model with parameter $\boldsymbol{\theta}_S$ over $\mathbb{D} \triangleq \cup_{k=1}^n \mathbb{D}^k$ with the help of a server in only one communication, as in

$$\min_{\boldsymbol{\theta}_S} \mathcal{L}(\theta_S) \triangleq \frac{1}{|\mathbb{D}|} \sum_{\{\mathbf{x}_i, y_i\} \in \mathbb{D}} \ell_{CE}(f_S(\mathbf{x}_i; \boldsymbol{\theta}_S), y_i), \tag{1}$$

where $\ell_{CE}(\cdot, \cdot)$ is the cross-entropy function, $f_S(\mathbf{x}_i; \boldsymbol{\theta}_S)$ is the prediction function of the server that outputs the logits (i.e., outputs of the last fully connected layer) of $\mathbf{x}_i$ given parameter $\boldsymbol{\theta}_S$.

Noticeably, in one-shot federated learning, the original training set $\mathbb{D}^k$ cannot be accessed, and only well-pretrained models parameterized by $\boldsymbol{\theta}_k$, are provided. Here, we define the ensemble as:

$$A_{\boldsymbol{w}}(\mathbf{x}; \{\boldsymbol{\theta}_k\}_{k=1}^n) \triangleq \sum_{k=1}^n w_k f_k(\mathbf{x}; \boldsymbol{\theta}_k), \tag{2}$$

where $f_k(\mathbf{x}; \boldsymbol{\theta}_k)$ denotes the prediction function that output the logits of $\mathbf{x}$ given $\boldsymbol{\theta}_k$, while $\boldsymbol{w} = [w_1, w_2, .., w_n]$ adjusts the weights of each local client logits. When $w_k = 1/n$, the ensemble is the same as the averaged ensemble, while when $w_k = n_k / \sum_{k=1}^{n} n_k$, the ensemble becomes weighted according to the data amount. For simplicity, in the rest paper, we use $A_{\boldsymbol{w}}$ to denote the ensemble and $A_{\boldsymbol{w}}(\mathbf{x})$ to denote $A_{\boldsymbol{w}}(\mathbf{x}; \{\boldsymbol{\theta}_k\}_{k=1}^{n})$, which means the output logits of the ensemble given $\mathbf{x}$.

When aggregating pre-trained models $\{\boldsymbol{\theta}_k\}_{k=1}^{n}$ into one server model $\boldsymbol{\theta}_S$, existing works mostly follow a two-stage framework. The first is to synthesize data $\mathbb{D}_S$ based on the ensemble output. In particular, giving a random noise $\boldsymbol{z}$ sampled from a standard Gaussian distribution and a random uniformly sampled label $y_s$, the generator $G(\cdot)$ with $\theta_G$ is responsible for generating the data $\mathbf{x}_S = G(\boldsymbol{z})$, forming the synthetic dataset $\mathbb{D}_S$. Typically, to make sure the synthetic data can be classified correctly with a high probability by the ensemble $A_{\boldsymbol{w}}$, the following loss is adopted:

$$\mathcal{L}(\boldsymbol{\theta}_G) \triangleq \frac{1}{|\mathbb{D}_S|} \sum_{\{\mathbf{x}_s, y_s\} \in \mathbb{D}_S} \ell_{CE}(A_{\boldsymbol{w}}(\mathbf{x}_s), y_s). \tag{3}$$

After getting the synthetic dataset $\mathbb{D}_S$ based on the generator in Eq.(3), OFL intends to distill the ensemble $A_{\boldsymbol{w}}$ into the final server model $\theta_S$ with the help of these synthetic data, as in:

$$\min_{\boldsymbol{\theta}_S} \mathcal{L}(\boldsymbol{\theta}_S) \triangleq \frac{1}{|\mathbb{D}_S|} \sum_{\{\mathbf{x}_s, y_s\} \in \mathbb{D}_S} \ell_{KL}(A_{\boldsymbol{w}}(\mathbf{x}_s), f_S(\mathbf{x}_s; \boldsymbol{\theta}_S)), \tag{4}$$

where $\ell_{KL}(\cdot, \cdot)$ denotes the Kullback-Leibler (KL) divergence.

Existing works illustrate that the performance of the server model is intrinsically related to the synthetic data $\mathbb{D}_S$ and the ensemble $A_{\boldsymbol{w}}$, which can also be concluded according to Eq.(4).

## 3.2 BOOSTING THE DATA QUALITY

Synthesizing data $\mathbb{D}_S$ is used to distill the ensemble model into the final server model as in Eq.(4). The quality of these synthesized data has been demonstrated vital to the distillation stage Lin et al. (2020). Moreover, since these data are also generated sourcing from the ensemble as in Eq.(3), it is of great importance to make these data embed as much the knowledge of the ensemble as possible and make them transferable to the final server model.

However, as hinted in Wang et al. (2020) and Zhang et al. (2022a), by utilizing only the CE loss, the synthesized data can be easily fitted by the server model, resulting in poor performance in the knowledge distillation stage. Therefore, to improve the quality of the generated data and make them focus more on transferable components, taking inspiration from Dong et al. (2020) and Li et al. (2023a), we increase the importance of hard samples while suppressing the importance of easy-to-fit samples in the generation stage. More specifically, given a prediction function $f$ which output logits, we employ the GHM introduced in Li et al. (2019) to measure the sample difficulty $d$ of $\mathbf{x}$:

$$d(\mathbf{x}, f) = 1 - \sigma(f(\mathbf{x}; \boldsymbol{\theta}))_y, \tag{5}$$

where $\sigma(f(\mathbf{x}; \boldsymbol{\theta}))_y$ is the probability on label $y$ predicted by the function $f(\cdot)$ with $\boldsymbol{\theta}$. Built upon the sample difficulty, we propose a hard-sample-enhanced loss $\mathcal{L}_H$ to synthesize data:

$$\mathcal{L}_H(\mathbf{x}_s, y_s; \boldsymbol{\theta}_G) \triangleq \frac{1}{|\mathbb{D}_S|} \sum_{\{\mathbf{x}_s, y_s\} \in \mathbb{D}_S} d(\mathbf{x}_s, A_{\boldsymbol{w}}) \ell_{CE}(A_{\boldsymbol{w}}(\mathbf{x}_s), y_s), \tag{6}$$

Moreover, to make synthesized samples hard for the server model to fit, an adversarial loss (Zhang et al., 2022c) is also introduced to generate hard samples. We try to maximize the differences in predictions between the ensemble model and the server model when generating data as follows:

$$\mathcal{L}_A(\mathbf{x}_s, \boldsymbol{\theta}_S; \boldsymbol{\theta}_G) \triangleq \frac{1}{|\mathbb{D}_S|} \sum_{\{\mathbf{x}_s, y_s\} \in \mathbb{D}_S} -\ell_{KL}(A_{\boldsymbol{w}}(\mathbf{x}_s), f_S(\mathbf{x}_s; \boldsymbol{\theta}_S)). \tag{7}$$

By combining the above losses, we can obtain the loss used to train the generator as follows:

$$\mathcal{L}(\boldsymbol{\theta}_G) \triangleq \mathcal{L}_H(\mathbf{x}_s, y_s; \boldsymbol{\theta}_G) + \beta \mathcal{L}_A(\mathbf{x}_s, \boldsymbol{\theta}_S; \boldsymbol{\theta}_G), \tag{8}$$

where $\beta$ is the scaling factor for the losses, which is set as 1 in the implementations.

Though samples synthesized using Eq.(8) are hard to fit for the current ensemble model, their difficulties for the server model are still lacking. This stems from the fact that the server model can easily fit these limited unchanged data during multiple distillation steps. To further promote the sample difficulty and diversity on the fly, we draw inspiration from adversarial learning (Goodfellow et al., 2014; Tashiro et al., 2020) to generate hard and diverse samples for the server model to learn.

More specifically, we diverse and increase the sample difficulty $d(\mathbf{x}_s, A_{\boldsymbol{w}})$ on the fly to make the synthetic samples hard to fit by introducing a perturbation $\boldsymbol{\delta}_i$ for each $\mathbf{x}_s$:

$$\boldsymbol{\delta}_i = \arg \max_{\|\boldsymbol{\delta}'\|_\infty \leq \epsilon} d(\mathbf{x}_s + \boldsymbol{\delta}', A_{\boldsymbol{w}}), \tag{9}$$

where $\|\cdot\|_\infty$ represents the $\mathcal{L}_\infty$-norm and $\epsilon$ controls the strength of perturbation. To simplify the computation and enhance the diversity, instead of the iterative adversarial attacks, we take only one step of loss backward to seek the direction in the input space that maximizes the similarity between the model output and a randomly sampled vector. The hard samples are constructed as follows:

$$\tilde{\mathbf{x}}_s \triangleq \mathbf{x}_s + \epsilon \frac{\nabla_{\mathbf{x}_s}(\boldsymbol{u}^\top A_{\boldsymbol{w}}(\mathbf{x}_s))}{\|\nabla_{\mathbf{x}_s}(\boldsymbol{u}^\top A_{\boldsymbol{w}}(\mathbf{x}_s))\|_2}, \tag{10}$$

where $\boldsymbol{u} \sim Unif([-1,1])^d$ is a randomly sample vector with dimension $d$. Following these, the originally generated hard samples are further harder and more diverse due to the randomness in $\boldsymbol{u}$.

By replacing each sample $x_s$ in $\mathbb{D}_S$ into $\tilde{\mathbf{x}}_s$, we can achieve a more hard and diverse synthetic dataset $\mathbb{D}_S$. Utilizing these hard samples, the knowledge of the ensemble model is transferred to the server model with parameter $\boldsymbol{\theta}_S$ by knowledge distillation the same as in Eq.(4).

Overall, with the hard sample technique embedding in the data synthesizing stage (replacing the generator loss in Eq.(3) with Eq.(8)) and making them diverse in the distillation stage (reconstruct the synthetic dataset $\mathbb{D}_S$ according to Eq.(10)), the quality of data generated and used for distillation becomes better, naturally boosting the performance of the server model.

### 3.3 BOOSTING THE ENSEMBLE QUALITY

The ensemble model takes the role of aggregating knowledge from all pre-trained models $\{\boldsymbol{\theta}_k\}_{k=1}^n$ and forms a virtually best-performance teacher. A straightforward method is to obtain the global model by averaging the parameters of all client models (e.g. FedAvg (McMahan et al., 2017)). However, FedAvg may fail to deliver a good performance when data among clients are non-IID (Karimireddy et al., 2020; Acar et al., 2021) and cannot handle the challenge of client model heterogeneity. Recent works (Heinbaugh et al., 2023; Diao et al., 2023) intend to construct a better ensemble model by altering the local training phase of each client, which may be unreliable, especially in today's model market scenarios. To tackle the client model heterogeneity and make the ensemble more practical, Guha et al. (2019); Zhang et al. (2022a) utilizes the direct ensemble $A_{\boldsymbol{w}}$ with $w_k = 1/n$ as the teacher, which means averaging the logits or weighted averaging them according to the number of the client data ($w_k = n_k / \sum_{k=1}^n n_k$). However, as suggested by Zhang et al. (2023) and Wang et al. (2023), the simple averaging or weighted averaging based on the client data amount may not be effective, especially in non-IID settings. There exists a better weighted combination of each client's contribution. Yet, their methods either need to alter the local training or transmit additional information, therefore their methods cannot be applied to one-shot FL.

To this end, we propose to boost the ensemble quality by searching for a more effective weighted ensemble of logits. As demonstrated by our experimental results in Fig. 1(b), given high-quality data (validation data), we can achieve a better ensemble with weights different from simple averaging or data amount based averaging. Fortunately, instead of using auxiliary data, we actually can acquire high-quality generated data from the hard synthesized samples set $\mathbb{D}_S$. Therefore, to get the best weights $\boldsymbol{w} = [w_1, w_2, \cdots, w_N]$ on $\mathbb{D}_S$, we need to solve the following optimization problem:

$$\min_{\boldsymbol{w}} \mathcal{L}_{\boldsymbol{w}}(\boldsymbol{w}) \triangleq \frac{1}{|\mathbb{D}_S|} \sum_{\{\mathbf{x}_s, y_s\} \in \mathbb{D}_S} \ell_{CE}(\sum_{k=1}^N w_k f_k(\mathbf{x}_s; \boldsymbol{\theta}_k), y_s), \tag{11}$$

where $y_s$ is the corresponding label to each synthesized hard samples $\mathbf{x}_s$. Exploring the optimal $\boldsymbol{w}$ requires multiple inner steps, leading to the training time to increase exponentially. Also inspired by

---

**Algorithm 1** Co-Boosting

---

1: **Input:** Clients' local models $\{\boldsymbol{\theta_1}, \cdots, \boldsymbol{\theta_n}\}$, server model $\boldsymbol{\theta}_S$, synthetic dataset $\mathbb{D}_S = \emptyset$, ensemble $A_{\boldsymbol{w}}$, generator $\boldsymbol{\theta}_G$, perturbation strength $\epsilon$, step size $\mu$, learning rate of generator and server $\eta_G$ and $\eta_S$, generation iterations $T_G$, global model training epochs $T$, and batch size $b$
2: **Output:** Global server model $\boldsymbol{\theta}_S$
3: **for** epoch = 0 to $T - 1$ **do**
4:     *// Generate hard synthetic samples*
5:     Sample a batch of noises and labels $\{\boldsymbol{z}_i, y_i\}_{i=1}^b$
6:     **for** $t_g = 0$ to $T_G - 1$ **do**
7:         Generate $\{\mathbf{x}_s\}_{i=1}^b$ with $\{\boldsymbol{z}_i\}_{i=1}^b$ and $\boldsymbol{\theta}_G$
8:         Update the generator: $\boldsymbol{\theta}_G \leftarrow \boldsymbol{\theta}_G - \eta_G \bigtriangledown_{\boldsymbol{\theta}_G} \mathcal{L}(\boldsymbol{\theta}_G)$, where $\mathcal{L}(\boldsymbol{\theta}_G)$ is defined in Eq.(8)
9:     **end for**
10:    $\mathbb{D}_S \leftarrow \mathbb{D}_S \cup \{\mathbf{x}_s\}_{i=1}^b$
11:    Diverse each sample $\{\mathbf{x}_s\}$ in $\mathbb{D}_S$ to $\{\tilde{\mathbf{x}}_s\}$ according to Eq.(10)
12:    *// Obtain a better ensemble*
13:    Update the mixing weights with $\mathbb{D}_S$ according to Eq.(12)
14:    Construct an updated ensemble $A_{\boldsymbol{w}}$ with updated $\boldsymbol{w}$ according to Eq.(2)
15:    *// Obtain the final server model*
16:    **for** sampling batch $\{\mathbf{x}_s\}$ in $\mathbb{D}_S$ **do**
17:        Update the server model: $\boldsymbol{\theta}_S \leftarrow \boldsymbol{\theta}_S - \eta_S \bigtriangledown_{\boldsymbol{\theta}_S} \mathcal{L}(\boldsymbol{\theta}_S)$, where $\mathcal{L}(\boldsymbol{\theta}_S)$ is defined in Eq.(4)
18:    **end for**
19: **end for**

---

methods of adversarial attacks (Goodfellow et al., 2014), we use the gradient's direction and fixed step size $\mu$ to update $\boldsymbol{w}$ every time after getting each batch of the synthesized data $\mathbb{D}_S$:

$$\boldsymbol{w}^t = \text{Normalize}(\boldsymbol{w}^{t-1} - \mu \text{sign}(\bigtriangledown_{\boldsymbol{w}} \mathcal{L}_{\boldsymbol{w}}(\boldsymbol{w}))), \tag{12}$$

where $\text{Normalize}$ denotes bounding each $w_k$ into $[0, 1]$ and $\text{sign}(\cdot)$ means the sign function.

The reweighting of each client's logit results in a superior ensemble model, which will naturally benefit the server model. Moreover, since the operations are done on the logit layer, this reweighting technique can be easily applied to both heterogeneous and homogeneous client model settings.

## 3.4 CO-BOOSTING THE DATA AND THE ENSEMBLE

As aforementioned, we introduce how to boost the data quality by utilizing hard sample techniques with a fixed ensemble and how to boost the ensemble with a fixed synthetic dataset. Actually, these two stages are inherently entangled and can boost each other at the same time. To get high-quality ensemble $A_{\boldsymbol{w}}$ and synthesized data $\mathbb{D}_S$, we are in fact trying to solve the following problem:

$$\min_{\boldsymbol{w}} \frac{1}{|\mathbb{D}_S|} \sum_{\{\mathbf{x}_s, y_s\} \in \mathbb{D}_S} \max_{\delta \in S} \ell_{CE}(\sum_{k=1}^n w_k f_k(\mathbf{x}_s + \delta; \boldsymbol{\theta}_k), y_s), \tag{13}$$

where $y_s$ is the label of sample $\mathbf{x}_s$, $\delta$ is the perturbation constrained in $S$. This problem can be addressed adversarially, which means the improvement of the data and the ensemble can be done simultaneously. With better quality data, the weighted ensemble can reach higher performance, while with this better ensemble, the data synthesized sourcing from this ensemble can further embed more knowledge. Therefore, by mutually boosting the quality of the synthesized data and the ensemble, we can naturally get a better-performance server model through the distillation in Eq.(4).

The overall algorithm is summarized in Algorithm 1. In each epoch, we first generate hard samples based on the current ensemble model and the last epoch server model. With these generated data, an enhanced ensemble model is cultivated by searching for the optimal ensembling weights of each client's logits. Utilizing the generated data and the upgraded ensemble, the final server model is trained by distilling the ensemble on these data. As illustrated in Sec. 3.2 and Sec. 3.3, with either one fixed, one can benefit from the other, thus by mutually boosting each other in the proposed Co-Boosting, we can achieve both better quality data and the ensemble periodically. Therefore, the global server model trained on them will inherently become better than other methods.

Table 1: Test accuracy of the server model of different methods over five datasets and across three levels of statistical heterogeneity (lower $\alpha$ is more heterogeneous).

| Method | $\alpha$ | FedAvg | FedDF | F-ADI | F-DAFL | DENSE | Co-Boosting |
|---|---|---|---|---|---|---|---|
| MNIST | 0.05 | 46.35±0.98 | 80.73±1.08 | 80.12±1.76 | 78.49±1.36 | 81.06±1.12 | **93.93±0.69** |
| | 0.1 | 75.68±0.82 | 87.91±0.92 | 85.92±0.82 | 87.44±0.61 | 87.83±1.38 | **94.44±1.02** |
| | 0.3 | 78.97±0.82 | **97.66±0.10** | 96.34±0.83 | 96.36±0.98 | 96.96±0.53 | 97.25±0.44 |
| FMNIST | 0.05 | 20.07±1.98 | 44.73±0.40 | 42.25±2.01 | 41.66±0.83 | 44.77±1.87 | **50.62±1.13** |
| | 0.1 | 46.61±1.94 | 68.40±1.80 | 63.19±1.99 | 67.81±0.93 | 69.43±1.94 | **74.86±1.70** |
| | 0.3 | 60.13±2.62 | **83.14±0.47** | 74.80±1.72 | 78.68±0.49 | 81.31±0.83 | 83.11±1.28 |
| SVHN | 0.05 | 39.41±2.19 | 60.79±0.52 | 56.58±1.05 | 59.38±1.19 | 60.24±1.31 | **65.40±0.86** |
| | 0.1 | 46.22±1.92 | 68.98±0.63 | 66.33±1.69 | 67.77±0.34 | 68.30±1.01 | **72.88±1.19** |
| | 0.3 | 72.61±2.06 | 79.78±0.55 | 76.75±1.47 | 78.01±1.02 | 78.73±0.84 | **81.31±1.09** |
| CIFAR-10 | 0.05 | 17.49±2.51 | 37.53±0.67 | 36.94±1.70 | 37.82±1.30 | 38.37±1.08 | **47.20±0.81** |
| | 0.1 | 27.54±1.80 | 49.63±0.80 | 47.19±0.97 | 46.32±0.97 | 47.80±1.21 | **57.09±0.94** |
| | 0.3 | 46.39±2.37 | 67.18±0.60 | 60.60±1.32 | 65.89±1.69 | 66.77±1.55 | **70.24±1.56** |
| CIFAR-100 | 0.05 | 6.45±0.92 | 16.07±0.54 | 13.75±1.01 | 15.79±0.21 | 16.17±1.33 | **19.24±1.42** |
| | 0.1 | 10.28±1.70 | 22.07±0.43 | 19.44±1.66 | 20.99±1.17 | 22.21±1.41 | **23.59±1.27** |
| | 0.3 | 15.22±2.08 | 30.71±0.53 | 26.14±1.37 | 28.79±1.25 | 30.33±1.24 | **31.30±1.30** |

# 4 EXPERIMENTS

## 4.1 EXPERIMENTAL DETAILS

**Datasets and partitions.** We conduct experiments on five real-world image datasets that are standard in the FL literature: MNIST (LeCun et al., 1998), FMNIST (Xiao et al., 2017), SVHN (Netzer et al., 2011), CIFAR10, and CIFAR100 (Krizhevsky et al., 2009). To simulate statistical heterogeneity, we use Dirichlet distribution to generate disjoint non-IID client training datasets as in Zhang et al. (2022a) and Heinbaugh et al. (2023). In particular, we sample $p_k \sim Dir(\alpha)$ and allocate a $p_k^i$ proportion of the data of class $i$ to client $k$. The parameter $\alpha$ controls the level of statistical imbalance, with a smaller $\alpha$ inducing more skewed label distributions among clients.

**Baselines.** Within the contemporary model-market scenarios, we compare the performance of Co-Boosting[1] against two existing methods: FedAvg (McMahan et al., 2017) and DENSE (Zhang et al., 2022a). Similar in (Zhang et al., 2022a), we also introduce two prevailing data-free KD methods DAFL (Chen et al., 2019) and ADI (Yin et al., 2020) and apply these to one-shot FL, giving F-DAFL and F-ADI. We also include FedDF (Lin et al., 2020) using the real validation dataset as the baseline.

**Configurations.** Following McMahan et al. (2017), we use CNN with 5 layers for SVHN, CIFAR10, and CIFAR100, LeNet-5 (LeCun et al., 1998) for MNIST and FMNIST. All available test data is used to evaluate the final server model (or ensemble). Unless otherwise stated, experiments are done with 10 clients and $Dir(0.1)$-parted. Results are reported averaged across at least 3 random seeds.

## 4.2 GENERAL RESULTS

**Overall Comparison.** To evaluate the effectiveness of our method, we conduct experiments under various non-IID settings by varying $\alpha = \{0.05, 0.1, 0.3\}$ and report the performance across different datasets and methods in Table 1. Notice, that we use the validation set in FedDF, which is not practical in the real world. From the table, we can conclude that Co-Boosting consistently outperforms all other baselines in all settings. Notably, in many settings, Co-Boosting achieves over a 5% accuracy improvement compared to the best baseline, DENSE. In cases of extreme statistical heterogeneity, such as when $\alpha = 0.05$, Co-Boosting surpasses the best baseline by substantial margins with 12.87%, 5.85%, 5.16%, 8.83%, and 3.07% on MNIST, FMNIST, SVHN, CIFAR-10, and CIFAR-100, respectively. We also compare the performance of the ensemble used in different methods on SVHN and CIFAR-10 in Table 2, others please refer to the Appendix. FedENS denotes averaged ensemble. The superior performance of the server model can be attributed to the enhanced

---

[1]Code is available at https://github.com/rong-dai/Co-Boosting

Table 2: Test accuracy of the ensemble in SVHN, and CIFAR-10 in $Dir$-parted setting.

| Dataset | SVHN | | | CIFAR-10 | | |
|---|---|---|---|---|---|---|
| Method | $\alpha$=0.05 | $\alpha$=0.1 | $\alpha$=0.3 | $\alpha$=0.05 | $\alpha$=0.1 | $\alpha$=0.3 |
| FedENS | 61.62±1.61 | 69.71±0.68 | 80.54±0.57 | 42.34±0.67 | 49.99±0.85 | 69.61±0.50 |
| Co-Boosting | **65.69±1.48** | **73.52±1.71** | **82.90±1.35** | **48.75±1.25** | **59.86±1.76** | **72.67±1.27** |

quality of synthesized data and the ensemble. While all compared methods, except FedAvg, utilize the direct logits averaging ensemble as the ensemble and aim to distill knowledge from it in a data-free manner, Co-Boosting, with its co-enhancing technique, results in a superior ensemble teacher, surpassing FedENS significantly. In a word, the superiority of our proposed method can be owed to the enhanced data and ensemble quality, which naturally translates into a better server model.

**Adaptation to Client Model Heterogeneity.** To evaluate our proposed method in a potential client heterogeneity setting, we apply five different models under a CIFAR-10, $Dir(0.1)$-parted setting. The heterogeneous models include CNN1 in McMahan et al. (2017), CNN2 in the pytorch tutorial (Paszke et al., 2019), ResNet (He et al., 2016), MobileNet (Howard et al., 2019), and ShuffleNet (Ma et al., 2018). Table 3 demonstrates the results of comparable methods, where Local denotes directly taking the pre-trained model for testing. We take ResNet as the server architecture and omit FedAvg as it does not support this setting. We use the same optimization hyperparameter for all methods and across all model architectures. We remark that as suggested in Zhang et al. (2022a) and Diao et al. (2021), FL under both non-IID data and different model architecture is a quite challenging task. Even under this setting, our proposed Co-Boosting still consistently outperforms other baselines by a large margin thanks to the benefits of making the ensemble and data improve together.

Table 3: Test accuracy of the server model architectured in RestNet in CIFAR-10 across three levels of statistical heterogeneity under a heterogeneous client model setting.

| $Dir(\cdot)$ | Local | FedDF | F-ADI | F-DAFL | DENSE | Co-Boosting |
|---|---|---|---|---|---|---|
| $\alpha$=0.05 | 46.49±1.97 | 56.51±1.87 | 55.64±1.96 | 55.14±1.51 | 55.79±1.75 | **58.64±1.83** |
| $\alpha$=0.1 | 51.70±1.14 | 59.49±1.06 | 60.30±1.70 | 58.58±1.49 | 58.67±0.88 | **62.30±0.90** |
| $\alpha$=0.3 | 67.61±1.54 | 70.53±1.19 | 71.61±1.24 | 71.92±1.95 | 72.98±1.61 | **75.02±1.37** |

## 4.3 IN-DEPTH STUDY

**Different Local Data Amount.** To further assess the effectiveness of Co-Boosting, which involves a better-weighted ensemble, we conduct experiments in an unbalanced local data setting. Similar to Acar et al. (2021), we sample data amounts for each client from a lognormal distribution. Higher values of $\sigma$ result in more unequal data distribution. Table 4 and Fig. 2 display the performance of the ensemble and the server, where the prefix 'DW-' signifies weighted averaging based on the local data amount. As observed, reweighting clients based on their data amount yields some benefits, but it falls short of achieving the best ensemble. Furthermore, when using the averaged ensemble FedENS as the teacher, all baseline methods perform poorly due to the suboptimal teacher. In contrast, benefiting from simultaneously boosting data and ensemble, the ensemble we get consistently outperforms FedENS. This leads to a substantial performance gain of the server model achieved in Co-Boosting over all baselines, with a margin of at least 10%.

Table 4: Test accuracy of ensemble.

| Method | $\sigma$=0.4 | $\sigma$=0.8 | $\sigma$=1.2 |
|---|---|---|---|
| FedENS | 46.87±1.02 | 41.86±1.20 | 37.88±1.38 |
| DW-FedENS | 47.80±1.21 | 53.25±0.52 | 47.52±0.40 |
| Co-Boosting | **58.94±0.50** | **57.41±1.12** | **55.27±1.72** |

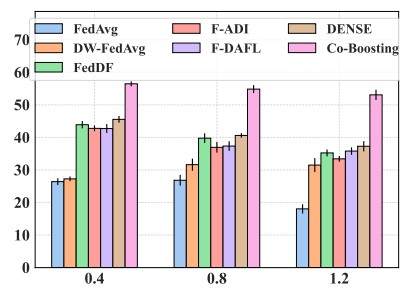

Figure 2: Test accuracy of server

**Different Data Distribution Shift.** Following Diao et al. (2023), we also conduct experiments on a $C\_cls$ partition setting, which means each client only possesses data of $C$ out of all classes. Results in Table 5 further demonstrate the superiority of our proposed method. With better data quality and better ensemble, our method consistently achieves the best server model.

Table 5: Test accuracy of the server of different methods in CIFAR-10 under $C\_cls$-parted setting.

| $C\_cls$ | FedAvg | FedDF | F-ADI | F-DAFL | DENSE | Co-Boosting |
|---|---|---|---|---|---|---|
| 2 | 16.15±2.61 | 23.07±0.83 | 23.16±1.12 | 24.53±0.57 | 23.85±0.93 | **36.37±1.85** |
| 3 | 26.47±1.46 | 38.39±0.64 | 36.06±1.93 | 38.13±1.17 | 38.14±1.38 | **53.91±1.80** |
| 4 | 33.78±2.42 | 54.51±1.12 | 51.04±1.40 | 52.53±0.97 | 51.53±1.79 | **58.00±1.59** |
| 5 | 35.95±1.96 | 58.34±1.58 | 55.27±2.06 | 54.67±1.26 | 56.79±1.03 | **62.52±1.75** |

**Different Number of Clients.** We evaluate the performance of these methods by varying the number of clients participating in OFL in Table 6. From the table, the final sever model still achieves the best accuracy when increasing the number of clients. This again validates that the increment quality of the ensemble model and data naturally brings a better server model.

Table 6: Test accuracy of the server model in CIFAR-10 across different numbers of clients.

| $n$ | FedAvg | FedDF | F-ADI | F-DAFL | DENSE | Co-Boosting |
|---|---|---|---|---|---|---|
| 5 | 36.94±1.74 | 50.62±0.98 | 48.76±0.78 | 49.76±1.42 | 50.53±1.02 | **54.29±1.38** |
| 10 | 27.54±1.80 | 49.63±0.80 | 47.19±0.97 | 46.32±0.97 | 47.80±1.21 | **57.09±0.94** |
| 20 | 26.34±1.97 | 38.98±0.99 | 38.93±0.64 | 36.28±1.39 | 38.86±0.42 | **49.56±0.98** |
| 50 | 23.01±0.94 | 29.52±0.62 | 27.45±1.13 | 29.41±0.90 | 28.51±0.54 | **42.29±1.43** |

**Effects of the proposed components.** We further study the effectiveness of our proposed hard sample generation loss in Sec.3.2, on-the-fly sample difficulty promotion in Sec.3.2, and ensemble enhancing in Sec.3.3. Table 7 shows the experimental results on SVHN and CIFAR-10 in a 10-client $Dir(0.05)$ parted setting. The results in the table illustrate that individually improving either data or ensemble leads to noticeable enhancements in the final server model performance. However, the most remarkable results are achieved when both data quality and ensemble capability are improved simultaneously. This finding strongly aligns with the underlying motivation of our study.

Table 7: Ablations on different components of our method. "GHS" for hard sample generation in the generator loss, "DHS" for on-the-fly diverse hard sample creation, and "EE" for ensemble enhancement through reweighting.

| GHS | DHS | EE | SVHN | CIFAR-10 |
|---|---|---|---|---|
| | | | 58.46 | 39.72 |
| ✔ | | | 61.18 | 42.85 |
| | ✔ | | 61.38 | 43.75 |
| | | ✔ | 62.67 | 41.45 |
| | ✔ | ✔ | 63.42 | 45.81 |
| ✔ | ✔ | | 62.46 | 44.36 |
| ✔ | | ✔ | 64.40 | 46.74 |
| ✔ | ✔ | ✔ | **65.40** | **47.20** |

**More facets.** For a thorough and comprehensive understanding, we operate sensitivity analyses of hyperparameters, compare with multi-round federated learning, and conduct experiments with heavier local models. Please refer to the Appendix for the results. Moreover, since our Co-Boosting needs no alternation of the local training, it can be combined with advanced local training. The results attached in the Appendix further demonstrate the superiority of our proposed Co-Boosting.

**Limitation.** The mixing weights are determined using synthetic samples. Though promising, there is still some disparity when compared to a mixing-weighted ensemble trained on real training data, as in Fig. 1(b). One possible way is to introduce virtual data (Tang et al., 2022). The exploration of methods to generate data capable of bridging this gap remains an avenue for further research.

## 5 CONCLUSION

In this paper, we seek to tackle the inherent bottleneck of one-shot federated learning, where the performance of the server model is inextricably linked with the quality of the generated data and the ensemble. We propose Co-Boosting, a novel method that facilitates a mutually beneficial relationship between data generation and ensemble improvement. By iteratively generating hard samples from the ensemble and enhancing the ensemble based on these data, Co-Boosting adversarially improves the quality of both the data and the ensemble, leading to the natural refinement of the server model. Extensive experiments across various settings validate the efficacy of our method and demonstrate that our method can be practically applied to contemporary model-market scenarios.

## ACKNOWLEDGEMENTS

RD, YGZ and BH were supported by the NSFC General Program No. 62376235, Guangdong Basic and Applied Basic Research Foundation No. 2022A1515011652, CCF-Baidu Open Fund, HKBU Faculty Niche Research Areas No. RC-FNRA-IG/22-23/SCI/04, and HKBU CSD Departmental Incentive Scheme. RD and XY were supported by National Natural Science Foundation of China (NSFC) under Grant U22A2094. TL is partially supported by the following Australian Research Council projects: FT220100318, DP220102121, LP220100527, LP220200949, IC190100031.

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

## A   MORE DISCUSSIONS ABOUT RELATED WORKS

### A.1   ONE-SHOT FEDERATED LEARNING

Large-scale data is beneficial to service providers in areas such as computer vision (Chang et al., 2023b;a), natural language processing (Peng et al., 2023), or video processing (Yang et al., 2021; 2022; Zhou et al., 2023; Song et al., 2023) among many others who can improve AI-based products or user experience via deep learning techniques with large data requirements. With the growing concern of data privacy, federated learning (FL) (McMahan et al., 2017) allows multiple decentralized clients to collectively train a machine learning model without the need to gather all clients' data. Most FL algorithms follow the communication prototype of FedAvg (McMahan et al., 2017), which requires many communication rounds to train an effective global. To minimize communication costs and mitigate potential security risks associated with multi-round communications, one-shot federated learning (OFL) has emerged as a promising research direction.

Compared to the straightforward baseline of parameter averaging, as in FedAvg (McMahan et al., 2017), Guha et al. (2019) is the first to propose OFL and utilizes the ensemble of each client's pre-trained models as the ensemble teacher. Subsequently, an auxiliary dataset is employed to distill knowledge from this ensemble and consolidate it into a unified server model. This paradigm, which is the mainstream of OFL, inherently relates the performance of the server to the data and ensemble used in the knowledge distillation stage.

With the purpose of using high-quality data, Li et al. (2021) proposes a two-tier knowledge distillation stage with the public dataset. Instead of using public data, Zhou et al. (2020) proposes to distill the local dataset on the client side and then send it to the server, which may help the aggregation. Yang et al. (2023) proposes to transmit class prototype and utilize auxiliary pre-trained diffusion models. To avoid transmitting additional information, Zhang et al. (2022a) proposes to generate fake data scouring from the direct ensemble and then use these fake data to distill the server model.

Regarding the improvement of the ensemble, Dennis et al. (2021) reforms the OFL task by utilizing a cluster-based method and requires uploading the cluster means. Diao et al. (2023) modify the local training phase of each client by introducing the placeholders in each client model. Heinbaugh et al. (2023) alters the local model into conditional variation auto-encoders and uses these to generate samples for ensemble and knowledge distillation. Nevertheless, it's worth noting that none of the previously mentioned approaches tackle enhancements in both data and ensemble simultaneously. Furthermore, only a few methods are applicable in real-world scenarios, particularly in the context of modern model-market scenarios (Vartak et al., 2016), where the server is supplied with exclusively well-pretrained models. In this context, there are constraints that include maintaining the integrity of the client's local training process, avoiding additional data transmissions, and accommodating potential variations in client model heterogeneity.

We compare these OFL algorithms in Table 8. Noticeably, we are the first to simultaneously enhance both data and ensemble, and our approach is adaptable to contemporary model-market scenarios.

Table 8: Comparison among existing one-shot FL algorithms.

| Method | Purpose | | Limitations of contemporary model-market scenarios | | | |
|---|---|---|---|---|---|---|
| | Improve data | Improve ensemble | No support of extra data | No alternations of local training | No additional transmission | Adapt to model heterogeneity |
| (Guha et al., 2019) | | | ✗ | ✔ | ✔ | ✗ |
| (Zhou et al., 2020) | ✔ | | ✔ | ✔ | ✗ | ✗ |
| (Li et al., 2021) | ✔ | | ✗ | ✗ | ✔ | ✔ |
| (Dennis et al., 2021) | | ✔ | ✔ | ✗ | ✗ | ✗ |
| (Zhang et al., 2022a) | ✔ | | ✔ | ✔ | ✔ | ✔ |
| (Diao et al., 2023) | | ✔ | ✔ | ✗ | ✔ | ✔ |
| (Heinbaugh et al., 2023) | | ✔ | ✔ | ✗ | ✗ | ✔ |
| (Yang et al., 2023) | ✔ | | ✗ | ✗ | ✗ | ✔ |
| Co-Boosting (ours) | ✔ | ✔ | ✔ | ✔ | ✔ | ✔ |

## A.2 KNOWLEDGE DISTILLATION

Knowledge distillation (KD) (Hinton et al., 2015) is proposed to transfer knowledge from one or more networks (teacher) to another (student). Typically, the student model is trained by minimizing the discrepancy between student and teacher logits generated using a suitable auxiliary dataset; KL-divergence is often chosen as the measure of the discrepancy.

In the same vein, Knowledge Distillation (KD) in federated learning aims to transfer knowledge from multiple local clients to a global server model. This concept was initially introduced by Lin et al. (2020) which proposed using KD at the server side based on an unlabeled auxiliary dataset. To reduce dependency on proxy datasets, researchers have turned to data-free knowledge distillation methods used in centralized settings, such as those proposed in Chen et al. (2019) and Yin et al. (2020). In the centralized setting, a generator or noise is optimized to mimic the behavior of the teacher model. These synthetic data are then used for knowledge distillation. Inspired by these centralized data-free KD works, generators are updated locally and aggregated globally to synthesize distillation samples, as demonstrated in the works Zhu et al. (2021) and Zhang et al. (2022b). Wang et al. (2023) further improved upon basic ensemble distillation by implementing weighted averaging based on locally trained discriminators.

However, in one-shot federated learning (OFL), conducting multiple rounds of training or transmitting generators and discriminators is not practical. Moreover, the requirement for an additional local client component contradicts the constraints in modern model-market OFL settings. A more serious concern is that a locally trained generator has direct access to the training samples, which could potentially lead to privacy leakage as it has the ability to remember all the training data Liu et al. (2019). In contrast, the generator in OFL is trained without access to even a single raw data point.

## A.3 MORE DISCUSSIONS

The parallels and contrasts with centralized co-learning techniques, such as those discussed in Qiao et al. (2018), are indeed intriguing to explore. It's important to note that centralized co-learning typically operates under the premise of learning from data that presents different views. However, in the one-shot federated learning context, the divergence in data distributions among clients is significantly more pronounced. Moreover, in OFL, models provided to the server are already pre-trained, which precludes the possibility of retraining local models. This distinction is crucial for understanding the unique challenges and approaches in OFL.

Additionally, viewing OFL through the lens of today's pre-trained foundation model era, as highlighted in works like Chen et al. (2022); Zhuang et al. (2023) opens up new avenues of exploration. Our work positions itself as a pioneering effort in adapting OFL to the era of foundation models. The key strength of our method is its independence from alterations in local training and its architecture-agnostic nature. This flexibility is particularly relevant in the context of foundation models, which are becoming increasingly central in various domains.

## B MORE EXPERIMENTS

### B.1 MORE DETAILS ABOUT THE EXPERIMENT CONFIGURATION

**Dataset.** We use five datasets in our experiments. MNIST (LeCun et al., 1998) is a large database of 10-class handwritten digits. It consists of 60,000 training images and 10,000 testing images, each of which is a 28x28 grayscale image. FMNIST (Xiao et al., 2017) is a dataset of Zalando's 10-class article images. It consists of a training set of 60,000 examples and a test set of 10,000 examples, each of which is a 28x28 grayscale image. SVHN (Netzer et al., 2011) is a real-world image dataset in 10 classes with a size of 32x32 used for developing machine learning and object recognition algorithms, 72,357 for train, and 26,032 for test. CIFAR-10 (Krizhevsky et al., 2009) consists of 60,000 32x32 color images in 10 classes, 50,000 for train, and 10,000 for test, while CIFAR-100 dataset is similar to the CIFAR-10 dataset but it has 100 classes containing 600 images each. Each image is also a 32x32 color image. For the FedDF method, we use 20% of the training set as a validation set for distillation. While all of the test data is used for testing.

**Baselines.** To accommodate the contemporary model-market scenarios as seen in Table 8, we compare the performance of Co-Boosting against two existing methods: FedAvg (McMahan et al., 2017) and DENSE (Zhang et al., 2022a). To ensure fair comparisons, we omit comparisons with methods that require the use of auxiliary public datasets, such as Li et al. (2021), or the modification of the local training phases of each client, as seen in Diao et al. (2023) and Heinbaugh et al. (2023). Both of these approaches may not be applicable in real-world scenarios. Moreover, since there is only one communication round, standard FL algorithms based on regularization terms including FedProx (Li et al., 2020b), SCAFFOLD (Karimireddy et al., 2020), and FedDyn (Acar et al., 2021) have no effect. Furthermore, since data-free one-shot FL can be addressed simply by operating data-free KD methods on the direct ensemble model of all models, similar in (Zhang et al., 2022a), we also introduce two prevalent data-free KD methods DAFL (Chen et al., 2019) and ADI (Yin et al., 2020) resulting in F-DAFL and F-ADI. More specifically, F-DAFL and F-ADI aim to either optimize a generator or a noise to mimic the performance of the ensemble teacher model. The data generated from this process is then utilized to distill knowledge into the student (server) model. Additionally, we include FedDF (Lin et al., 2020) using the validation dataset.

**Hyper-parameters.** Unless otherwise stated, we conduct experiments with 10 clients and Dir(0.1) (high heterogeneous) partition. Results are reported averaged across at least 3 random seeds. For each client's local training, we use the SGD optimizer with momentum=0.9 and learning rate=0.01. We set the batch size to 128 and the local epoch to 300. Following McMahan et al. (2017), we use a simple CNN with 5 layers for SVHN, CIFAR10, and CIFAR100, LeNet-5 (LeCun et al., 1998) for MNIST and FMNIST. All available test data is used to evaluate the final server model (or ensemble). The generator we use is the same as in Zhang et al. (2022a); Chen et al. (2019) and it is trained by Adam optimizer with a learning rate $\eta_g = 1e3$ over $T_G = 30$ rounds. The distillation temperature in the knowledge distillation stage used in the server model stage is set to 4, while the temperature used in the KL loss in the generator loss is set to 1. The perturbation strength is set to $\epsilon = 8/255$ and the step size $\mu$ is set to $0.1/n$. For the training of the server model $f_S(\cdot)$, we use the SGD optimizer with learning rate $\eta_S = 0.01$ and momentum=0.9. The number of total epochs $T$ is set to 500. We maintain the same hyperparameters for all baseline methods.

## B.2 Comparison of the Ensemble Model

Our Co-Boosting method, in addition to improving the capabilities of the final server model, also yields an enhanced ensemble model through iterative enhancement of synthetic data and ensemble learning, as illustrated in Algorithm 1. The server model obtained through Co-Boosting can be considered as distilled from the progressively improved ensemble and the generated hard samples. Therefore, investigating the performance of this ensemble is also highly meaningful. In this section, we present the final ensembles obtained by Co-Boosting under various experimental settings corresponding to the main paper and compare them with the ensembles used by other methods, FedENS (average logit output). Results on $Dir$-parted setting, $C\_cls$-parted setting, and different numbers of client settings are demonstrated in Table 9, 10 and 11 respectively.

Table 9: Test accuracy of the ensemble in MNIST, FMNIST, and CIFAR-100 in $Dir$-parted setting.

| Dataset | MNIST | | | FMNIST | | | CIFAR-100 | | |
|---|---|---|---|---|---|---|---|---|---|
| Method | $\alpha$=0.05 | $\alpha$=0.1 | $\alpha$=0.3 | $\alpha$=0.05 | $\alpha$=0.1 | $\alpha$=0.3 | $\alpha$=0.05 | $\alpha$=0.1 | $\alpha$=0.3 |
| FedENS | 81.52±0.89 | 88.16±1.27 | 97.75±0.15 | 45.51±1.05 | 71.34±0.66 | 83.56±1.24 | 21.39±0.52 | 26.56±1.35 | 35.56±1.01 |
| Co-Boosting | **94.59±0.79** | **94.47±0.74** | **97.75±0.27** | **52.43±1.93** | **76.52±2.08** | **83.11±1.28** | **22.57±0.99** | **27.71±1.68** | **36.29±1.24** |

Table 10: Test accuracy of the ensemble in CIFAR-10 in the $C\_cls$ partition setting.

| Method | $C$=2 | $C$=3 | $C$=4 | $C$=5 |
|---|---|---|---|---|
| FedENS | 24.43±1.62 | 39.67±1.03 | 56.37±1.40 | 60.01±1.46 |
| Co-Boosting | **37.77±1.13** | **56.25±1.22** | **61.80±1.34** | **64.96±1.23** |

Table 11: Test accuracy of the ensemble in CIFAR-10 across different numbers of clients.

| Method | n=5 | n=10 | n=20 | n=50 |
|---|---|---|---|---|
| FedENS | 53.27±0.66 | 49.99±0.85 | 43.47±1.42 | 32.60±1.22 |
| Co-Boosting | **58.80±1.15** | **59.86±1.76** | **51.14±0.86** | **45.05±1.08** |

As can be easily concluded from the above tables, the final ensemble model we get in our proposed method consistently outperforms the baseline FedENS to a large extent. This demonstrates that by finding an optimal weight based on the hard samples we generate, we can form a much better ensemble, which will naturally in turn become a better teacher in the server distillation age. It is noteworthy that in cases where the data partition exhibits a higher degree of difficulty (resulting in greater distribution shift among clients), the weighted ensemble we obtain can outperform the baseline weighted average by a more substantial margin. This phenomenon may be attributed to the fact that when clients share a more similar distribution of data, their knowledge tends to converge, making simple averaging somewhat sufficient to encapsulate their collective knowledge. Nevertheless, this is still not optimal. Our weighted ensemble, learned based on the hard samples, excels in determining more effective weights for aggregating all clients across all settings.

## B.3 MORE EXPERIMENTS ON MORE DATASETS

To comprehensively evaluate our method's performance, we conduct experiments using widely used Tiny-ImageNet dataset (Le & Yang, 2015) (200 classes, 500 training samples each) with ResNet18 serving as the client architecture backbone. Using the same settings as other experiments, we explore server models based on both CNN and ResNet18 architectures. The results, presented in Table 12, not only confirm the effectiveness of our method but also highlight its architecture-agnostic feature.

Table 12: Test accuracy of the server model of different methods in Tiny-Imagenet dataset over two architectures and across three levels of statistical heterogeneity (lower $\alpha$ is more heterogeneous).

| Server | $\alpha$ | FedAvg | FedDF | F-ADI | F-DAFL | DENSE | Co-Boosting |
|--------|----------|--------|-------|-------|--------|-------|-------------|
| CNN | 0.05 | - | 6.78±0.13 | 6.44±0.78 | 6.90±0.08 | 6.71±0.14 | **7.38±0.08** |
| | 0.1 | - | 9.71±0.20 | 9.57±0.71 | 9.77±0.48 | 9.09±0.43 | **10.10±0.69** |
| | 0.3 | - | 12.62±0.45 | 11.94±0.83 | 12.65±0.54 | 12.71±0.72 | **13.83±0.84** |
| ResNet | 0.05 | 0.70±0.20 | 5.65±0.30 | 5.70±0.38 | 5.68±0.50 | 5.92±0.18 | **8.21±0.12** |
| | 0.1 | 1.19±0.84 | 8.63±0.82 | 7.01±0.26 | 7.81±0.22 | 8.88±0.23 | **10.29±0.43** |
| | 0.3 | 2.09±0.52 | 11.93±0.38 | 11.58±0.84 | 12.30±0.36 | 13.05±0.36 | **14.35±0.93** |

It is also of great importance to evaluate our proposed Co-Boosting under scenarios involving feature shifts. To this end, we extend our experiments to include domain generalization datasets, specifically MNIST-M (Ganin et al., 2016) and PACS Li et al. (2017). Experiments follow the settings outlined in Li et al. (2023b); Zhang et al. (2023), which focus on federated domain generalization. In these experiments, each domain is allocated to a different client, and we employ the leave-one-domain-out testing technique. We utilized CNN for MNIST-M and ResNet18 for PACS. To further evaluate our method's versatility, we tested with both CNN and ResNet18 as server architectures. Results in Table 13 and Table 14 demonstrate the superior performance of our proposed method, which we attribute to the mutual enhancement principle inherent in our approach. The results not only validates our method's model-agnostic feature but also its effectiveness in settings with feature distribution shifts.

Table 13: Test accuracy of the server model of different methods in MNIST-M dataset over two architectures on the unseen domain using leave-one-out test mechanism.

| Server | Domain | FedAvg | FedDF | F-ADI | F-DAFL | DENSE | Co-Boosting |
|--------|--------|--------|-------|-------|--------|-------|-------------|
| CNN | MNIST | 25.82±1.40 | 66.38±1.37 | 66.27±1.48 | 66.75±1.13 | 70.81±1.12 | **82.15±0.83** |
| | MNIST-M | 16.40±1.61 | 41.18±0.34 | 39.34±1.50 | 39.85±1.02 | 41.35±1.19 | **42.85±0.80** |
| | SVHN | 18.88±1.72 | 47.91±1.78 | 45.38±1.86 | 45.05±0.83 | 46.20±1.10 | **53.23±1.67** |
| | SYN | 45.97±1.96 | 78.91±0.29 | 78.70±0.57 | 79.83±0.55 | **79.90±0.30** | 78.67±0.69 |
| | AVG | 26.77±1.67 | 58.59±1.12 | 57.39±1.35 | 57.87±0.88 | 59.57±0.93 | **64.23±1.00** |
| ResNet | MNIST | - | 72.68±1.53 | 70.91±1.23 | 69.15±2.06 | 73.25±1.59 | **84.39±2.31** |
| | MNIST-M | - | 43.42±0.54 | 42.40±1.22 | 41.69±1.34 | 43.71±1.56 | **45.25±1.03** |
| | SVHN | - | 47.91±0.88 | 48.02±0.70 | 48.46±1.48 | 47.27±1.19 | **53.75±1.13** |
| | SYN | - | 80.48±0.51 | 79.49±1.54 | **80.55±0.88** | 80.03±0.44 | 77.93±1.22 |
| | AVG | - | 61.12±0.86 | 60.21±1.17 | 59.96±1.44 | 61.07±1.20 | **65.33±1.42** |

Table 14: Test accuracy of the server model of different methods in PACS dataset over two architectures on the unseen domain using leave-one-out test mechanism.

| Server | Domain | FedAvg | FedDF | F-ADI | F-DAFL | DENSE | Co-Boosting |
|---|---|---|---|---|---|---|---|
| CNN | P | - | 33.80±0.45 | 32.71±1.53 | 34.25±2.12 | 35.68±1.83 | **50.77±1.78** |
| | A | - | 22.56±0.43 | 21.17±0.52 | 21.14±0.78 | 22.41±0.39 | **23.54±1.18** |
| | C | - | 29.65±0.30 | 29.32±0.83 | 29.61±0.86 | 29.01±0.99 | **37.59±1.41** |
| | S | - | 30.45±0.49 | 29.85±0.26 | 28.36±0.76 | 30.17±0.34 | **30.67±2.67** |
| | AVG | - | 29.12±0.42 | 28.26±0.79 | 28.34±1.13 | 29.34±0.89 | **35.64±1.66** |
| ResNet | P | 11.32±1.93 | 37.09±0.45 | 36.50±1.04 | 37.55±1.78 | 37.19±1.89 | **51.43±1.72** |
| | A | 18.50±2.88 | 23.76±0.85 | 22.95±0.82 | 22.92±1.89 | 24.83±1.49 | **26.76±1.23** |
| | C | 16.60±1.28 | 29.62±0.48 | 28.53±0.89 | 29.35±1.68 | 31.78±1.30 | **36.73±1.24** |
| | S | 19.65±2.63 | 30.20±0.42 | 29.77±0.85 | 29.67±0.76 | 32.00±1.20 | **35.35±1.74** |
| | AVG | 16.52±2.18 | 30.16±0.55 | 29.31±0.90 | 29.87±1.53 | 31.45±1.47 | **37.56±1.48** |

## B.4 MORE EXPERIMENTS IN LOCAL MODEL HETEROGENEITY SETTING

Similar to the main paper, we evaluated the performance of the proposed Co-Boosting against other baselines under the same client model heterogeneity setting, while varying the server model architecture. Results are shown in Table 15, all experiments are done in a 5-client $Dir(0.1)$ parted setting. While it is natural to observe variations in performance among different architectures, this can be attributed to the use of the same hyperparameters across all architectures and the inherent differences in representation capacity among them. However, it is noteworthy that Co-Boosting consistently outperforms other baselines regardless of the server architecture. This further underscores the effectiveness of the proposed Co-Boosting technique for enhancing both data and ensemble.

Table 15: Test accuracy of server model in different architectures in CIFAR-10 across three levels of statistical heterogeneity under a heterogeneous client model setting.

| $\alpha$ | Architecture | Local | FedDF | F-ADI | F-DAFL | DENSE | Co-Boosting |
|---|---|---|---|---|---|---|---|
| 0.05 | CNN1 | 41.06±1.49 | 52.96±1.66 | 51.69±1.39 | 51.73±0.90 | 52.66±1.43 | **56.10±1.64** |
| | CNN2 | 30.03±0.69 | 44.09±0.51 | 43.38±1.73 | 43.13±1.61 | 43.82±1.49 | **45.82±1.36** |
| | MobileNet | 37.32±1.52 | 49.47±1.93 | 50.55±1.31 | 49.48±0.66 | 50.35±1.33 | **52.97±1.18** |
| | ShuffleNet | 38.09±1.77 | 51.05±0.23 | 48.75±0.68 | 49.95±1.17 | 50.01±1.98 | **53.45±1.93** |
| 0.1 | CNN1 | 48.05±1.68 | 57.66±0.76 | 56.18±1.67 | 57.04±1.54 | 58.31±1.21 | **61.87±1.86** |
| | CNN2 | 39.90±0.25 | 46.21±1.36 | 43.03±1.56 | 45.56±1.94 | 45.88±2.32 | **48.31±1.60** |
| | MobileNet | 44.72±1.39 | 50.25±1.04 | 49.36±1.44 | 51.91±0.98 | 51.47±0.52 | **54.75±1.58** |
| | ShuffleNet | 47.12±1.71 | 53.15±1.68 | 52.88±1.41 | 52.04±1.36 | 52.25±1.00 | **55.72±0.77** |
| 0.3 | CNN1 | 60.16±0.75 | 67.08±0.86 | 68.94±0.92 | 67.55±1.22 | 67.89±1.08 | **70.06±0.92** |
| | CNN2 | 51.79±0.75 | 58.52±1.41 | 57.49±0.86 | 57.43±1.19 | 58.15±1.53 | **62.45±1.58** |
| | MobileNet | 55.54±1.31 | 60.60±0.38 | 60.42±1.07 | 60.11±1.12 | 61.18±0.84 | **64.20±1.07** |
| | ShuffleNet | 55.49±0.80 | 61.82±0.89 | 60.30±0.73 | 59.98±1.93 | 61.19±1.24 | **63.22±1.19** |

## B.5 COMBINATION WITH ADVANCED LOCAL TRAINING

To alleviate the non-IID data problem in the Federated Learning (FL) setting, recent works (Qu et al., 2022; Caldarola et al., 2022; Dai et al., 2023) have shown that the use of the Sharpness Aware Minimization (SAM) technique can be beneficial. It is suggested that incorporating this technique into local training can lead to improved server aggregation. In this section, to investigate the performance of methods utilizing SAM for local training in the one-shot federated learning setup, we conduct experiments on SVHN and CIFAR-10 datasets. The experiments are done with 10 clients under a Dir(0.05) partition scenario. From Table 16, it is evident that introducing SAM-based techniques during the local training phase, as compared to the baseline in Table 1, indeed enhances the performance of various methods. This improvement is attributed to the potential alignment achieved during local training, which ultimately leads to better server performance upon aggregation. While, our proposed method, Co-Boosting, continues to outperform all the baselines, highlighting the su-

periority of our approach. Moreover, this shows that our method is not dependent on the specifics of the local training phase. With a better local training method, our approach can further improve.

Table 16: Test accuracy of the server of different methods when combining advanced local training.

| Dataset | FedAvg | FedDF | F-ADI | F-DAFL | DENSE | Co-Boosting |
|---|---|---|---|---|---|---|
| SVHN | 43.46±3.18 | 62.67±1.12 | 56.94±1.35 | 61.18±1.59 | 62.00±1.47 | **68.34±1.39** |
| CIFAR-10 | 23.39±3.93 | 39.54±1.69 | 37.88±1.83 | 38.53±0.60 | 41.14±1.65 | **49.04±0.73** |

## B.6 MORE EXPERIMENTS WITH HEAVIER MODELS

In this section, we conduct further experiments using larger models, VGG11 (Simonyan & Zisserman, 2014) and ResNet18 (He et al., 2016), on CIFAR-10. The experiments were carried out with 10 clients in a federated setting partitioned according to a $Dir(0.05)$ distribution, which represents a significantly large difference in data distribution. Results in Table 17 show that our method continues to perform well with heavier models. Our proposed Co-Boosting outperforms the existing baselines by a significant margin, approximately 10% in both cases of VGG and ResNet architectures.

Table 17: Test accuracy of server model with VGG and ResNet in CIFAR-10.

| Model | FedAvg | FedDF | F-ADI | F-DAFL | DENSE | Co-Boosting |
|---|---|---|---|---|---|---|
| VGG | 34.90±2.70 | 52.92±0.98 | 52.28±1.80 | 52.84±1.29 | 53.32±1.58 | **63.37±1.29** |
| ResNet | 33.55±3.59 | 52.62±0.93 | 51.53±1.95 | 52.65±1.18 | 52.98±1.81 | **62.18±1.92** |

## B.7 SENSITIVITY ANALYSIS OF HYPERPARAMETERS

**Step size $\mu$.** $\mu$ is the step size used when adjusting the ensemble weights of each client. It controls the range of weight changes based on the synthetic data. As can be seen from Table 18, when $\mu$ is very small, the performance will decline because the range of weight exploration is not large. When $\mu$ is of moderate size, it does not have a significant impact on the final ensemble and the final server model. This is because after each weight adjustment, a normalization operation is performed to limit it within a certain range. Overall, we recommend using $0.1/n$ as the step size.

Table 18: Analysis of $\mu$ under a 10-client $Dir(0.1)$-parted setting in CIFAR-10.

| | 0.005 | 0.01 | 0.05 | 0.1 |
|---|---|---|---|---|
| Ensemble | 59.13±1.38 | 59.86±1.76 | 59.18±1.93 | 59.20±1.04 |
| Server | 56.46±1.79 | 57.09±0.94 | 57.44±2.01 | 57.52±1.20 |

**Perturbation strength $\epsilon$.** $\epsilon$ is the parameter that adjusts the difficulty of synthesizing data on the fly during data distillation. It controls the range of pixel-level changes on the image plane. As can be seen from Table 19, when $\epsilon$ is small, it cannot generate sufficiently diverse hard samples, thus not maximizing the extraction of information from the samples. Conversely, when the $\epsilon$ is large, it causes significant damage at the image level, potentially destroying semantic information. Therefore, we recommend using a moderate epsilon value. In this paper, we use 8/255 as the default.

Table 19: Analysis of $\epsilon$ under a 10-client $Dir(0.05)$-parted setting in SVHN.

| 1/255 | 2/255 | 4/255 | 8/255 | 16/255 | 32/255 |
|---|---|---|---|---|---|
| 61.97±1.23 | 63.53±0.93 | 64.71±1.20 | 65.40±0.86 | 65.58±1.20 | 64.25±1.32 |

### B.8 COMPARISON WITH MULTI-ROUND FEDERATED LEARNING

In this section, we run FedAvg (McMahan et al., 2017), and Scaffold (Karimireddy et al., 2020), two popular and efficient algorithms for 200 communication rounds. There are 10 clients with 10 local epochs in each round. Fig. 3 shows that our approach can outperform the multi-round FedAvg algorithm across various settings. Furthermore, our method remains competitive with the results of the Scaffold algorithm, which undergoes 200 rounds of communication. However, it's crucial to note that the performance improvement in multi-round algorithms is a result of multiple rounds of information exchange and model alignment. The multi-round paradigm entails the possibility of client dropouts and vulnerability to potential attacks. In contrast, our algorithm necessitates only a single communication round, making it exceptionally suitable for real-world applications.

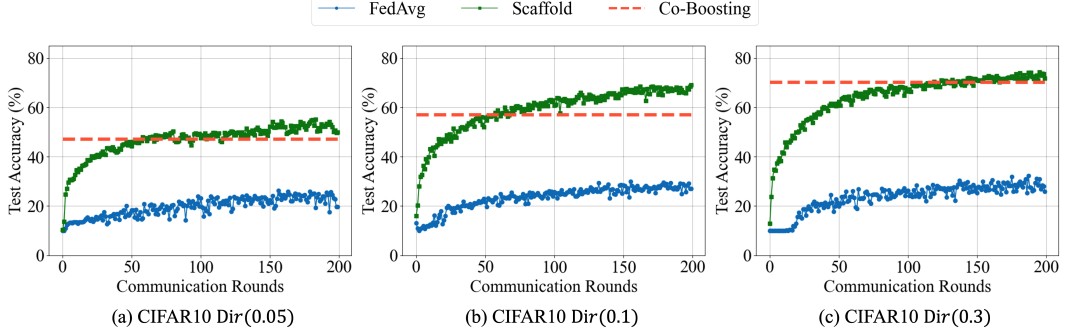

Figure 3: Comparison with multi-round federated learning methods.

### B.9 VISUALIZATION OF SYNTHETIC DATA

We conduct data visualization for the synthetic data of our proposed Co-Boosting on the MNIST and CIFAR-10 datasets in Fig. 4 and Fig. 5, where digits 0 to 9 represent 10 different classes. There are a total of five rows corresponding to images generated at the 100th, 200th, 300th, 400th, and 500th global model training epochs. For each epoch, we randomly select 3 images for display. From the figures, it can be observed that the generated data does not hold significant practical value to the human eye. However, it is sufficient for machine learning models to learn a robust classifier.

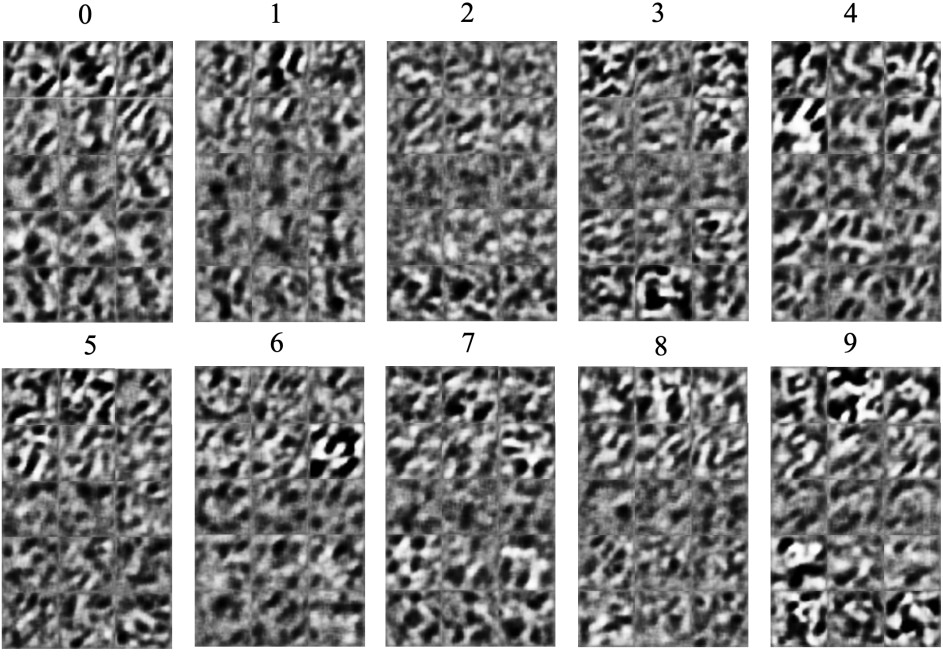

Figure 4: Visualization of synthetic data on MNIST dataset.

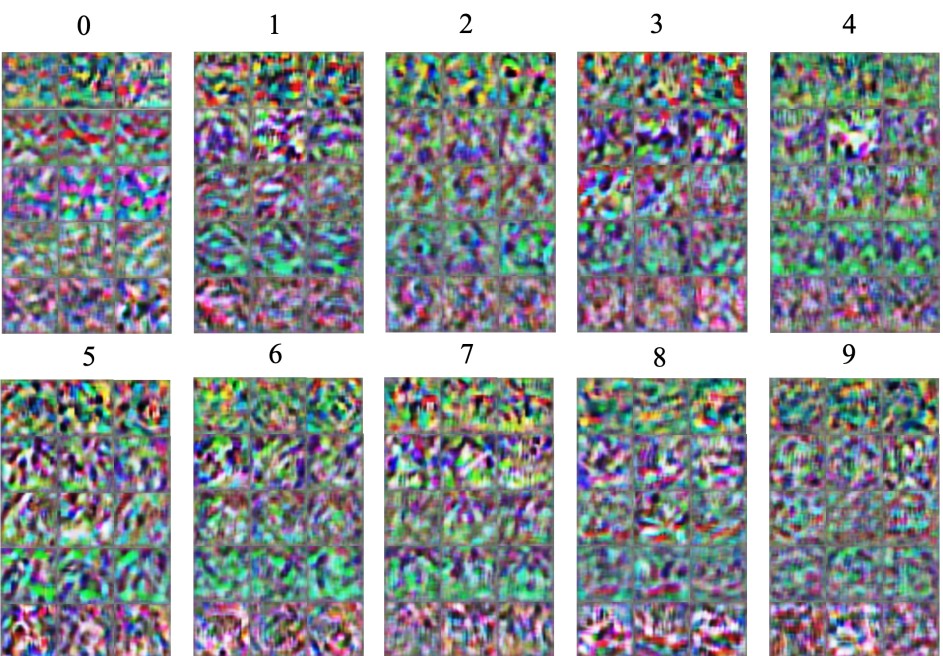

Figure 5: Visualization of synthetic data on CIFAR-10 dataset.

