# OpenReview forum: "Enhancing One-Shot Federated Learning Through Data and Ensemble Co-Boosting"
_ICLR.cc/2024/Conference — ICLR 2024 poster_

### Official Review · Reviewer_rouU · 2023-10-27

**Soundness:** 2 fair
**Presentation:** 2 fair
**Contribution:** 2 fair
**Rating:** 6
**Confidence:** 3

**Summary:**

The paper proposes a method for one-shot federated learning named "co-boosting" that aims to improve ensemble and data quality for training the server model. The main contribution lies in the utilization of adversarial samples for training the server-side model.

**Strengths:**

On a high level, the idea of OFL and co-boosting is understandable if one wants to avoid multi-round FL then the method is useful. The evaluation is thorough across datasets, architectures, baselines and varying federated settings. From results it looks like method performs exceptionally well compare to the baselines at least on MNIST, FMNIST and SVHN.

**Weaknesses:**

The important details of synthetic data generation and ensemble creation are vague and overly complicated, which makes the paper very hard to follow and identify the key components of the method. As a result, it is difficult to judge if the method is indeed OFL. The evaluation is thorough, but if it is not conducted when the clients' models are trained only once, then I have serious concerns about what the paper is claiming and how the experiments are conducted.

Section 3.2 is extremely hard to follow and lacks structure. It is unclear whether the data is generated on the fly during the training of the server model or if it is pregenerated.

Algorithm 1 does not provide any useful information beyond what has already been described in earlier sections. Please highlight what happens on the server and what happens on the client device. If it is truly OFL, what does the epoch correspond to? Is it the server model training? This question also relates to Figure 1 (a) and (d). If there are multiple epochs of both the client and server, how is it considered OFL?

**Questions:**

"Though samples synthesized using Eq.(8) are hard to fit for the current ensemble model, their difficulties for the server model are still lacking". If the synthesized samples are hard to fit by the ensemble, why make it even more harder for the server model to learn by introducing adversarial samples?

What kind of issues arise when one has multi-round FL? This question is related to the following statement: "mitigating errors arising from multi-round." Is OFL a silver bullet? when it can fail ? Please explain in Limitation section to better contextualize when using OFL is useful.

Where does the generator G(·) come from? What is the architecture, and how is it trained? It is completely unclear how data is generated for knowledge distillation eq. 4 to train the server model. I suggest the authors write a separate section explaining it in detail since it is a main component of the proposed method. Is it pure random noise? If yes, why not use [1]?

How does Co-Boosting not require model transmission? From eq. 2, it seems like the ensemble is created with FedAvg in a single round. But as per section 3.3, the ensemble is not created by weighted averaging of client models? Is that correct? So logits are averaged on synthetic data, which acts as an ensemble for the server model, what is correct?

What is GHM? Please provide the full form and a minimal description to improve the readability of the paper. There are already too many references/links to prior work and moving back and forth makes it hard to follow the method.  In eq. 5, does $\bf{x}$ represent a synthetic sample?

Regarding the statement "searching for the optimal ensembling weights of each client's logits," it is unclear how the search is performed.

[1] Baradad Jurjo, Manel, et al. "Learning to see by looking at noise." Advances in Neural Information Processing Systems 34 (2021): 2556-2569.

---

> ### Author Response · Authors · 2023-11-17
>
> We would like to thank the reviewer for taking the time to review our work. We appreciate that you find our experiments extensive and thorough. According to your valuable comments, we provide detailed feedback.
>
> **Q1**: Setting issues:
> >**a):** The important details of synthetic data generation and ensemble creation are vague and overly complicated, which makes the paper very hard to follow and identify the key components of the method. As a result, it is difficult to judge if the method is indeed OFL. The evaluation is thorough, but if it is not conducted when the clients' models are trained only once, then I have serious concerns about what the paper is claiming and how the experiments are conducted.
> >**b):** Section 3.2 is extremely hard to follow and lacks structure. It is unclear whether the data is generated on the fly during the training of the server model or if it is pregenerated.
> >**c):** Algorithm 1 does not provide any useful information beyond what has already been described in earlier sections. Please highlight what happens on the server and what happens on the client device. If it is truly OFL, what does the epoch correspond to? Is it the server model training? This question also relates to Figure 1 (a) and (d). If there are multiple epochs of both the client and server, how is it considered OFL?
>
> ***Ans for Q1):***
> We apologize for the misunderstanding. In response to your valuable comments, we have added the following explanations to our revision.
>
> Replay to **a)** and **c)**: We have added more explanations for our method in the OFL context. To clarify, OFL involves aggregating information from multiple pre-trained client models with various architectures into a unified server model. In contemporary model market scenarios, all algorithms, including ours, operate exclusively on the server side. This means that there is **only one communication round between the server and clients**. Once the server acquires the client models, no further communication is needed, and all computations are performed on the server side. We would like to clarify that the term "epoch" in our paper refers solely to the training epoch of the server model, and all experiments adhere to an OFL setting.
>
> We would like to demonstrate the process of our proposed method as illustrated in Fig. 1a. In each training epoch, our algorithm starts by generating ‘hard’ samples based on the current state of the weighted ensemble and the server model (lines 4-11 in Algorithm 1). These challenging samples are then used to adjust the weights of the ensemble (lines 12-14 in Algorithm 1). The server model is subsequently updated by distilling knowledge from both the enriched data and the refined ensemble (lines 15-18 in Algorithm 1). This process represents one epoch and is not a singular enhancement but a continuous cycle where the quality of data and the ensemble are iteratively improved.
>
> Reply to **a)** and **b)**: Regarding the **ensemble, it is constructed as a weighted combination of each client's logit outputs**, as defined in Eq. (2). The ensemble comprises pre-trained models, and the only modifiable parameters are the weights of each client. This can be seen as a function of client weights outputting weighted logits, with the **pre-trained models remaining unchanged**. The data generation stage is detailed and discussed in section 3.2. Previous OFL methods generate data that can be classified by the ensemble, using the loss defined in Eq. (3), which is later used in the distillation stage (Eq. (4)) to transfer knowledge from the ensemble to the server. However, we identify that this loss (Eq. (3)) is insufficient for effective knowledge transfer. To address this, we introduce a hard sample emphasizing loss (Eqs. (6) and (7)). The generator is optimized according to the loss defined Eq. (8), providing the data for subsequent use (lines 5-10 in Algorithm 1). Additionally, Eqs. (9) and (10) are employed to dynamically make the generated data harder in each epoch (lines 10-11 in Algorithm 1).
>
> Reply to **b)**: The data is **generated on-the-fly** in each epoch for the server model's training, and it is not pre-generated. We would like to note that previous OFL methods focus on either improving the ensemble or improving the data quality. Once this step is achieved, knowledge is distilled from these static high-quality sources into the server model. This process is linear, where the enhancement of data or the ensemble is a prerequisite step before the server model can benefit. In contrast, our method adopts **an iterative and interconnected approach**, integrating high-quality data generation and ensemble refinement into a continuous, dynamic cycle. This novel approach to linking these two processes is a key contribution of our proposed paradigm, demonstrating its uniqueness and potential impact in the field of OFL.
>
> We hope these clarifications assist in better understanding the process and contributions of our proposed method.

---

> > ### Author Response · Authors · 2023-11-17
> >
> > **Q2**: Data generation issues:
> > > "Though samples synthesized using Eq.(8) are hard to fit for the current ensemble model, their difficulties for the server model are still lacking". If the synthesized samples are hard to fit by the ensemble, why make it even more harder for the server model to learn by introducing adversarial samples?
> >
> > ***Ans for Q2):***
> > Thank you for posing this insightful question. We have added more explanations as follows in the revised paper.
> >
> > We would like to first note the dynamics of our training process, which is periodic in nature. At each epoch, we generate examples, reweight clients based on these examples, and distill knowledge from the refined ensemble into the server model, utilizing all generated data up to that point. As per Eq. (8), while the data samples are indeed challenging for the ensemble, they could become less so for the server model over time. This is due to **the iterative nature of the learning process**: if the generated data remains unchanged across multiple epochs, there's a risk that the server model might overfit to this static set of hard samples.
> >
> > To **prevent this potential overfitting**, we designed our approach to dynamically diversify and increase the difficulty of samples for the server model in each epoch. This is achieved through the introduction of Eq. (10). By continuously presenting the server model with more challenging samples, we ensure that it doesn't simply learn to fit the initial set of hard samples, but rather evolves to tackle increasingly complex data.
> >
> > The effectiveness of this strategy noted in _DHS_ is demonstrated in our ablation studies, presented in Table 7. These studies show that the technique of generating progressively harder samples on-the-fly significantly contributes to the overall success of our proposed method.
> >
> > **Q3**: Multi-round FL comparsions:
> > > What kind of issues arise when one has multi-round FL? This question is related to the following statement: "mitigating errors arising from multi-round." Is OFL a silver bullet? when it can fail ? Please explain in Limitation section to better contextualize when using OFL is useful.
> >
> > ***Ans for Q3):***
> > This is an excellent question that invites a comparison between multi-round Federated Learning (FL) and One-shot Federated Learning (OFL), following your valuable comments, we have added more discussions as follows in the revised paper.
> >
> > As highlighted in previous OFL works [2]-[7], most FL algorithms traditionally adhere to the multi-communication paradigm exemplified by FedAvg. This paradigm requires numerous communication rounds to train an effective global model. However, several limitations are associated with the multi-round framework:
> >
> > - **Communication Overheads and Connectivity Issues**: There is a significant communication burden, along with the risk of connection drop errors between clients and the server, as detailed in [8][9]. These challenges can severely impact the efficiency and reliability of the FL process.
> >
> > - **Privacy and Security Concerns**: The multi-round approach poses potential risks for man-in-the-middle attacks [10], alongside various other privacy and security concerns [11]. These risks are particularly pertinent in scenarios where sensitive data is involved.
> >
> > Furthermore, in **contemporary model market scenarios** that involve multiple pre-trained and large-scale models, it becomes impractical to engage in repeated transmissions and retraining processes.
> >
> > While it is acknowledged that limiting transmissions to a single round might potentially impact overall performance, we believe that the study and development of OFL are warranted. OFL presents a promising area of research, particularly in contexts where minimizing communication overheads, enhancing privacy, and efficiently leveraging pre-trained models are critical.

---

> > > ### Author Response · Authors · 2023-11-17
> > >
> > > **Q4**: Generator issues:
> > > > Where does the generator G(·) come from? What is the architecture, and how is it trained? It is completely unclear how data is generated for knowledge distillation eq. 4 to train the server model. I suggest the authors write a separate section explaining it in detail since it is a main component of the proposed method. Is it pure random noise? If yes, why not use [1]
> > > >
> > > ***Ans for Q4):***
> > > Thank you for your insightful question and for suggesting further clarification. We apologize for any confusion caused and appreciate the opportunity to elaborate on the generator G(·) used in our method.
> > >
> > > One of the crux of our method lies in the synthetic data generated by the generator. This data is used for the knowledge distillation stage, where it plays a key role in training the server model. We adopt **a generator framework, consistent with prior work [7], which utilizes a simple deep convolutional network**. Detailed specifics of its architecture will be elaborated upon in the appendix.
> > >
> > > As outlined in Section 3.2 and Algorithm 1 (lines 5-9), the generator functionally transforms random noise inputs $z_i$ and corresponding assigned labels $y_i$ into synthetic data $x_i=G(z_i)$. The optimization of the generator, governed by the loss defined in Eq. (8), yields a refined generator $G^*()$ and its output $x_i=G^*(z_i)$ with label $y_i$. This generated data is stored for the subsequent knowledge distillation process.
> > >
> > > Post optimization, the generated samples **are not pure random noise**, but designed to be recognized correctly by the ensemble model while presenting a challenge for the server model to identify.
> > >
> > > Thank you for directing our attention to [1]. Inspired by your valuable comments, we indeed find it intriguing to conceptualize OFL as a process of learning from noise. While prior research in this area has predominantly focused on either rendering the generated noise more realistic or on enhancing the ensemble model, our approach represents a bridge between these two critical elements. We integrate the process of data generation with ensemble boosting, allowing them to mutually enhance each other.
> > >
> > > **Q5**: Ensemble issues:
> > > >How does Co-Boosting not require model transmission? From eq. 2, it seems like the ensemble is created with FedAvg in a single round. But as per section 3.3, the ensemble is not created by weighted averaging of client models? Is that correct? So logits are averaged on synthetic data, which acts as an ensemble for the server model, what is correct?
> > > >
> > > ***Ans for Q5):***
> > > We would like to note that the ensemble defined in Eq. 2 represents the weighted output logits of each client model. This ensemble can be conceptualized as a function that, given certain inputs, outputs weighted logits. The pre-trained client models within this ensemble remain unchanged throughout the process. The only modifiable parameters are the weights assigned to each client's logit output. This design lends our method its architecture-agnostic characteristic, allowing it to adapt to various model architectures seamlessly.
> > >
> > > In our approach, to enhance the capabilities of this ensemble model, we **adaptively learn the weights of these clients using Eqs. 11 and 12**. This learning process is conducted using the synthetic data generated by our system, while the pre-trained client models remain static during the entire procedure.
> > >
> > > In summary, our Co-Boosting method **does not require model transmission** because the ensemble is not created through weighted averaging of client models in the traditional sense. Instead, it involves adjusting the weights of the output logits of the pre-trained client models based on synthetic data. This process ensures that the client models remain unaltered, while the ensemble's effectiveness is continually improved through dynamic weight adjustments.
> > >
> > > **Q6**: Writing issues:
> > > >What is GHM? Please provide the full form and a minimal description to improve the readability of the paper. There are already too many references/links to prior work and moving back and forth makes it hard to follow the method. In eq. 5, does x represent a synthetic sample?
> > >
> > > ***Ans for Q6):***
> > > We apologize for any confusion caused by the use of the acronym GHM without a full explanation. GHM stands for **Gradient Harmonizing Mechanism**, a method introduced in [12] for assessing the difficulty of samples.
> > >
> > > To enhance the readability of our paper and to reduce the need for constant cross-referencing, we have **summarized the relevant related works, in Appendix A**. This section provides a comprehensive and detailed discussion of these methods, aiding in understanding their role in our study.
> > >
> > > Regarding Eq. 5, GHM is employed as a universal tool to gauge the difficulty of any image input. In this context, the variable x in Eq. 5 **represents a generic image input**, not limited to synthetic samples. The function d() within this equation is utilized to compute the difficulty of these samples.

---

> > > > ### Author Response · Authors · 2023-11-17
> > > >
> > > > **Q7**: Reweighting issues:
> > > > >Regarding the statement "searching for the optimal ensembling weights of each client's logits," it is unclear how the search is performed.
> > > > >
> > > > ***Ans for Q7):***
> > > > We appreciate your inquiry regarding the process of determining the optimal ensemble weights for each client's logits. Accordingly, we have added more explanations as follows in the revised paper.
> > > >
> > > > We would like to note that as defined in our method, the ensemble's tunable parameters are exclusively the weights assigned to each client's logit outputs. In each epoch of server training, after generating hard samples, our goal is to ascertain the most effective weights when applied to these synthetic data. To address this, we engage in solving the optimization problem outlined in Eq. 11. Rather than developing a bespoke optimizer for the client weights, we **employ a gradient descent method to fine-tune the ensembling weights**. This approach is straightforward yet efficient and is elaborated in detail in Eq. 12. Since there are multiple epochs for the server model training, we think this design strikes a balance between simplicity and effectiveness, allowing for the precise tuning of the ensemble weights without introducing unnecessary complexity into the optimization process.
> > > >
> > > > > ***Reference***
> > > > >
> > > > > [1] Learning to see by looking at noise. In NIPS2021.
> > > > >
> > > > > [2] One-shot federated learning. In ArXiv preprint 2019.
> > > > >
> > > > > [3] Practical one-shot federated learning for cross-silo setting. In IJCAI 2021.
> > > > >
> > > > > [4] Heterogeneity for the win: One-shot federated clustering. In ICML 2021.
> > > > >
> > > > > [5] Towards addressing label skews in one-shot federated learning. In ICLR 2023.
> > > > >
> > > > > [6] Data-free one-shot federated learning under very high statistical heterogeneity. In ICLR 2023.
> > > > >
> > > > > [7] Dense: Data-free one-shot federated learning. In NIPS 2022.
> > > > >
> > > > > [8] Federated learning: Challenges, methods, and future directions. In IEEE signal processing magazine 2020.
> > > > >
> > > > > [9] Advances and open problems in federated learning. In Foundations and Trends in Machine Learning 2021.
> > > > >
> > > > > [10] A survey on security and privacy of federated learning. In Future Generation Computer Systems 2021.
> > > > >
> > > > > [11] See through gradients: Image batch recovery via gradinversion. In CVPR 2021.
> > > > >
> > > > > [12] Gradient harmonized single-stage detector. In AAAI 2019.

---

> ### Author Response · Authors · 2023-11-20
> **Welcome for more discussions**
>
> Dear reviewer #rouU,
>
> Thanks for your valuable time in reviewing and insightful comments. Following your comments, we have tried our best to provide responses and revise our paper. Here is a **summary of our response** for your convenience:
>
> - (1) **Setting issues**: We apologize for the misunderstanding. In response to your valuable comments, we have added the following explanations to our revision. OFL means there is only one communication round between the server and clients. We would like to clarify that the term "epoch" in our paper refers solely to the training epoch of the server model, and all experiments adhere to an OFL setting. The ensemble is constructed as a weighted combination of each client's logit outputs with the inside pre-trained models remaining unchanged. Regarding the data, it is generated on-the-fly in each epoch for the server model's training. Detailed explanations of the generator design, learnable ensemble weights, and how our method work has been added to the revision to enhance our work.
> - (2) **Data generation issues**: Thank you for posing this insightful question. We would like to note that at each epoch, we generate examples, reweight clients based on these examples, and distill knowledge from the refined ensemble into the server model, utilizing all generated data up to that point. Due to the iterative nature of the learning process, there's a risk that the server model might overfit this static set of hard samples. Therefore, to prevent this potential overfitting, we designed our approach to dynamically diversify and increase the difficulty of samples for the server model in each epoch.
> - (3) **Multi-round FL comparisons:**: Following your valuable comments, we have added more discussions as follows in the revised paper. We would like to note that there may be communication overheads and connectivity issues in multi-round FL, which may severely impact the efficiency and reliability of the FL process. Moreover, the multi-round approach poses potential risks for man-in-the-middle attacks, alongside various other privacy and security concerns. Last but not least, in contemporary model market scenarios that involve multiple pre-trained and large-scale models, it becomes impractical to engage in repeated transmissions and retraining processes. We have added discussions in the revision to enhance our work.
>
> We humbly hope our response has addressed your concerns. If you have any additional concerns or comments that we may have missed in our responses, we would be most grateful for any further feedback from you to help us further enhance our work.
>
> Best regards
>
> Authors of #1615

---

> ### Author Response · Authors · 2023-11-20
> **Window for responsing and draft updating is closing**
>
> Dear Reviewer #rouU,
>
> Thanks very much for your time and valuable comments. We understand you're busy. But as the window for responding and paper revision is closing, would you mind checking our response ([a brief summary](https://openreview.net/forum?id=tm8s3696Ox&noteId=cabYIsDfXB), and [details](https://openreview.net/forum?id=tm8s3696Ox&noteId=ofa7gEiobH)) and confirm whether you have any further questions? We are very glad to provide answers and revisions to your further questions.
>
> Best regards and thanks,
>
> Authors of #1615

---

> ### Author Response · Authors · 2023-11-21
> **Window for responding and draft updating is closing**
>
> Dear Reviewer #rouU,
>
> Thanks very much for your time and valuable comments. We understand you're busy. But as the window for responding and paper revision is closing, would you mind checking our response ([a brief summary](https://openreview.net/forum?id=tm8s3696Ox&noteId=cabYIsDfXB), and [details](https://openreview.net/forum?id=tm8s3696Ox&noteId=ofa7gEiobH)) and confirm whether you have any further questions? We are very glad to provide answers and revisions to your further questions.
>
> Best regards and thanks,
>
> Authors of #1615

---

> ### Comment · Reviewer_rouU · 2023-11-21
>
> Thanks for the effort in providing explanation and making relevant changes. In summary, the method does the following: a) train locally on client data, generate logits on synthetic data, 3) server aggregates the logits to create so called ensemble (which does not sound like an ensemble in true sense), and then server applies knowledge distillation on data generated by G + optimizing on so called ensemble logits with GHM in the loop. Is it correct?

---

> > ### Author Response · Authors · 2023-11-21
> >
> > Dear Reviewer #rouU,
> >
> > Thank you for your thoughtful review and insightful comments. We appreciate your time and effort in understanding our method. We agree that the server model is aggregated by distilling knowledge from the ensemble on the synthetic data and the data is generated with hard sample technique including GHM defined in Eq.(6), adversarial loss defined in Eq.(7), and on the fly diversity technique defined in Eq.(10).
> >
> > To ensure complete clarity and address any potential misunderstandings, we would like to offer additional explanations. In OFL, clients train models locally on their data and transmit these models to the server. This transmission is a one-time process. With these pre-trained models which may potentially have different architectures, the challenge lies in training a new server model using these pre-trained models without accessing real data. A core aspect is synthesizing high-quality data and creating an effective 'teacher' (ensemble) for the final server model to operate knowledge distillation. In OFL, this ensemble is defined by the weighted logits from all pre-trained models when given data. We would like to note that the ensemble is characterized not by being bound to specific data but by functioning as a mechanism that inputs images and outputs weighted logits from all the contained, frozen pre-trained models. Prior works in OFL have focused either on enhancing ensemble capabilities or improving synthetic data quality. While our method works as follows:
> > - a) Sample Generation: (See lines 5-9 in Algorithm 1) We generate high-quality samples with G by using GHM (Eq.6) and adversarial loss (Eq.7).
> > - b) Sample Diversification: (Lines 10-11 in Algorithm 1) We further diversify these samples as per Eq.(10).
> > - c) Ensemble Update: (Lines 12-14 in Algorithm 1) We use the synthetic data to update the ensemble by reweighting each pre-trained model.
> > - d) Knowledge Distillation: (Lines 15-18 in Algorithm 1) We distill knowledge from the enhanced ensemble into the final server model with the help of the synthetic data.
> >
> > We would like to note that a) to d) is operated repeatedly which diverges our method from previous works in a way that periodically and mutually enhances both the ensemble and data quality.
> >
> > We hope this explanation provides a clearer understanding of our method and its distinction from previous approaches in OFL. We are happy to provide further details if needed.

---

> ### Comment · Reviewer_rouU · 2023-11-21
>
> "Ensemble Update: (Lines 12-14 in Algorithm 1) We use the synthetic data to update the ensemble by reweighting each pre-trained model." -> these are logits as per your earlier explanation! I suggest to stick to one definition.
>
> Thanks for the clarification. I am increasing the score but I am not very convinced that this method is proposing something entirely new in terms of methodological contributions to FL.

---

> > ### Author Response · Authors · 2023-11-21
> > **Thanks for increasing scores**
> >
> > Dear reviewer #rouU,
> >
> > Thanks for your swift reply despite such a busy period. We sincerely appreciate that you can raise the score. Sorry for any misunderstanding caused by our explanations. We would like to clarify that the ensemble functions as a mechanism that inputs images and outputs weighted logits from all the contained, frozen pre-trained models. So the tunable parameters are the weight of logits outputed by each pre-trained models.
> >
> > Regarding the contribution concern, we would like to note that the main contribution is the proposed co-boosting approach. It is the first time to make the ensemble and the synthetic data mutually boost each other. Also thanks to the inherent design, our method eliminates the need for modifications to the client's local training, requires no additional data or model transmission, and accommodates heterogeneous client model architectures. We provide a comprehensive comparison with previous methods and summarize them in Table 8.
> >
> > If you have any further concerns or comments that we may have inadvertently missed in our previous responses, we would sincerely appreciate any additional feedback from you. Your valuable comments will greatly assist us in enhancing our work. We extend our heartfelt gratitude for your swift response and increasing scores.
> >
> > Best regards and thanks,
> >
> > Authors of #1615

---

### Official Review · Reviewer_vHVL · 2023-10-30

**Soundness:** 2 fair
**Presentation:** 3 good
**Contribution:** 2 fair
**Rating:** 6
**Confidence:** 4

**Summary:**

This paper developed Co-Boosting, a new framework in which synthesized data and the ensemble model mutually enhance each other progressively. Extensive experiments demonstrated the good performance of the proposed method.

**Strengths:**

1. Develop a new framework that can enhance synthesized data and the ensemble model mutually
2. The generation of hard samples seems interesting and helpful.
3. Extensive experiments have shown the proposed method outperforms the baselines.

**Weaknesses:**

1. Fig 1 (a) needs to be elaborated to help readers better understand the 3 steps. Please elaborate on these steps in the caption.

2. Missing one-shot FL baselines. The introduction discussed two methods, Heinbaugh et al. (2023), and Diao et al. (2023). However, the authors did not compare them in the experiments.

3. How to evaluate whether the labels generated in the synthetic data are accurate and correct?

4. The technical contribution is not very significant. It seems to combine some existing works. For instance, generating hard samples motivated by existing work like Dong et al. (2020) and Li et al. (2023).

**Questions:**

How to evaluate whether the labels generated in the synthetic data are accurate and correct? Whether the pseudo label affects the results a lot?

---

> ### Author Response · Authors · 2023-11-17
>
> We would like to thank the reviewer for taking the time to review our work. We appreciate that you find our method instesesting and experiments are extensive. According to your valuable comments, we provide detailed feedback.
>
> **Q1**: Caption issues:
> > Fig 1 (a) needs to be elaborated to help readers better understand the 3 steps. Please elaborate on these steps in the caption.
> >
> ***Ans for Q1):***
> Thank you for your valuable suggestion regarding Fig. 1 (a). We have revised the figure caption to provide a more detailed explanation of the three key steps involved in each epoch of the server training process. The revised caption now reads as follows: In each epoch of server training, high-quality samples are first generated based on last epoch’s ensemble and server, which are then used to adjust client weights giving a better ensemble. Based on the enriched data and refined ensemble, server model is updated by distilling knowledge from them.
>
> **Q2**: Baseline issues:
> > Missing one-shot FL baselines. The introduction discussed two methods, Heinbaugh et al. (2023), and Diao et al. (2023). However, the authors did not compare them in the experiments.
> >
> ***Ans for Q2):***
> Thank you for bringing attention to OFL methods proposed by Diao et al. (2023) and Heinbaugh et al. (2023). Accordingly, we have added the following discussions in the revised paper.
>
> - The primary reason for this omission is grounded in the **practical constraints of contemporary model market scenarios**, which form the basis of our research approach in OFL. In contemporary settings, the server typically has access only to pre-trained client models, without the feasibility of imposing modifications on the local training phase of these models. The methods introduced by Diao et al. and Heinbaugh et al. both necessitate alterations to the local training process of client models. Diao et al. integrate placeholders in client models, while Heinbaugh et al. convert local models into conditional variational auto-encoders. Such modifications, while innovative, are **not always practical or feasible in real-world model market scenarios** where the server's interaction with client models is limited to the use of pre-trained, unmodifiable models.
>
> - Both the methods by Diao et al. and Heinbaugh et al. primarily focus on improving the ensemble model. Once an enhanced ensemble is obtained, it is used in a fixed state for knowledge distillation with additional data. We have outlined a detailed comparison with these two mentioned and other baselines, which are discussed in Appendix A.1 and presented in Table 8. The rationale behind our baseline selection is further elaborated in Appendix B.1. Notably, these previous baselines, including those by Diao et al. and Heinbaugh et al., concentrate on improving either the ensemble or the data. In contrast, our method innovatively proposes a mechanism that allows for the periodic mutual enhancement of both the ensemble and the data.
>
> Inspired by your valuable comments, it would be intriguing to investigate how our method performs when the constraint of fixed local training is relaxed, and alterations in client local training are permitted. To explore this, we have conducted preliminary experiments, presented in Appendix B.6, demonstrating that our method can **indeed benefit from advanced local training techniques**. These findings suggest the potential for further enhancing our approach by leveraging improvements in client model training.

---

> > ### Author Response · Authors · 2023-11-17
> >
> > **Q3**: Pseudo label issues:
> > > How to evaluate whether the labels generated in the synthetic data are accurate and correct? Whether the pseudo label affects the results a lot?
> > >
> > ***Ans for Q3):***
> > Thank you for your insightful inquiry regarding the accuracy and correctness of the pseudo labels used in our synthetic data generation process. Accordingly, we have added the following explanations to the revision as follows.
> >
> > The psedudo label is used in the data generation stage. For each random noise input, we manually assign a pseudo label and use the generator, along with the loss function detailed in Eq. 8, to optimize the output. Initially, given a random noise $z_i$ and an assigned label $y_i$, the generated sample are $\{G(z_i),y_i\}$. After the generator optimization, the sample transforms into $\{G^*(z_i),y_i\}$.
> >
> > The question of whether these labels are assigned accurately or correctly is indeed intriguing. In our experiments, we randomly assign a class label to each piece of random noise. To investigate the impact of these assigned labels, we conduct tests where labels are assigned in an orderly and balanced manner across batch noises. Interestingly, the results were similar to those obtained with randomly assigned labels as follows (experiments are done in CIFAR-10 Dir-parted). Based on these observations, we infer that the specific assignment of labels does not significantly affect the overall performance. This inference aligns with the design of our method; as per Eq. 8, the generator is optimized to produce samples that are likely to be correctly predicted by the ensemble yet remain challenging for the server model to recognize.
> >
> > **Table: In-depth studies on assigned labels**
> > | $\alpha$ | Random       | Order      |
> > |---------|--------|-------------|
> > | 0.05 | 47.20±0.81 | 46.19±1.00  |
> > | 0.1 | 57.09±0.94 | 57.14±0.59 |
> > | 0.3 | 70.24±1.56 | 70.83±1.64 |
> >
> > **Q4**: Contribution concern:
> > > The technical contribution is not very significant. It seems to combine some existing works. For instance, generating hard samples motivated by existing work like Dong et al. (2020) and Li et al. (2023).
> > >
> > ***Ans for Q4):***
> > We apologize for the misunderstanding. We would like to highlight our main contributions as follows.
> >
> > - We appreciate the reviewer's observation regarding the inspiration we drew from the hard sample generating techniques described in Dong et al. (2020) and Li et al. (2023). However, we wish to emphasize the novel contribution of our work, which goes beyond the these existing methods.
> >
> > - In the context of OFL, where the efficacy of the final server model hinges on the combined learning from an ensemble and generated data [1][2], our work addresses a unique and critical challenge: enhancing the capabilities of these two components in tandem. As detailed in Appendix A.1 and illustrated in Table 8, prior research predominantly focuses on either amplifying the ensemble or the data generation process in isolation. These approaches adopt a linear methodology, where once high-resource data or an enhanced ensemble is obtained, it is kept static during the server model training.
> >
> > - Different from previous works, our work pioneers an integrated approach, establishing **a dynamic and mutually reinforcing relationship between the ensemble and data generation processes**. This novel strategy enables the periodic improvement of both data quality and ensemble performance. By interlinking hard sample generation with our unique reweighting techniques, we facilitate a continuous cycle of enhancement for both data and ensemble. This synergy, we argue, is our main contribution, leading to a naturally evolved, better server model in OFL.
> >
> > > ***Reference***
> > >
> > > [1] One-shot federated learning. In ArXiv preprint 2019
> > >
> > > [2] Dense: Data-free one-shot federated learning. In NIPS 2022.

---

> > ### Comment · Reviewer_vHVL · 2023-11-22
> > **Thanks for your response**
> >
> > Thanks for your detailed response! I think your answers have addressed most of my concerns. I will raise the score.

---

> > > ### Author Response · Authors · 2023-11-23
> > > **Thanks for increasing scores**
> > >
> > > Dear reviewer #vHVL,
> > >
> > > Thank you for your prompt response and raising the score. If you have any further questions or comments, we are very glad to discuss more with you. Your valuable comments will greatly help us in enhancing our work.
> > >
> > > Best regards
> > >
> > > Authors of #1615

---

> ### Author Response · Authors · 2023-11-20
> **Welcome for more discussions**
>
> Dear reviewer #vHVL,
>
> Thanks for your valuable time in reviewing and insightful comments. Following your comments, we have tried our best to provide responses and revise our paper. Here is a **summary of our response** for your convenience:
>
> - (1) **Baseline issues**: Following your valuable comments, we have added the following discussions in the revised paper. The primary reason for this omission is grounded in the practical constraints of contemporary model market scenarios, which form the basis of our research approach in OFL. Detailed explanations of the baseline choosing are added to the revision. Also inspired by your valuable comments, we conduct experiments when the constraint of fixed local training is relaxed, and alterations in client local training are permitted. Results in Appendix B.6, demonstrate that our method can indeed benefit from advanced local training techniques.
> - (2) **Pseudo label correctness concern**: Thank you for your insightful inquiry. We conduct experiments on assigning the synthetic data orderly or random labels. Results in the following demonstrate that the specific assignment of labels does not significantly affect the overall performance. The assigned pseudo label is used to make the samples produced by the generator likely to be correctly predicted by the ensemble yet remains challenging for the server model to recognize.
> - (3) **Contribution concern**: We would like to note that the main contribution is the proposed co-boosting approach. It is **the first time** to make the ensemble and the synthetic data **mutually boost each other**. Existing literature focuses on either enhancing the ensemble or the data quality in isolation. Once a higher quality of data or an improved ensemble is achieved, these elements typically remain static during the server training phase. Our work diverges from this norm by proposing a method that establishes **a dynamic and mutually reinforcing relationship between the ensemble and data generation processes.** We provide a comprehensive comparison and summarize them in Table 8 in response.
>
> We humbly hope our response has addressed your concerns. If you have any additional concerns or comments that we may have missed in our responses, we would be most grateful for any further feedback from you to help us further enhance our work.
>
> Best regards
>
> Authors of #1615

---

> ### Author Response · Authors · 2023-11-20
> **Window for responsing and draft updating is closing**
>
> Dear Reviewer #vHVL,
>
> Thanks very much for your time and valuable comments. We understand you're busy. But as the window for responding and paper revision is closing, would you mind checking our response ([a brief summary](https://openreview.net/forum?id=tm8s3696Ox&noteId=IxCcQNinhB), and [details](https://openreview.net/forum?id=tm8s3696Ox&noteId=uohlmGQYuZ)) and confirm whether you have any further questions? We are very glad to provide answers and revisions to your further questions.
>
> Best regards and thanks,
>
> Authors of #1615

---

> ### Author Response · Authors · 2023-11-21
> **Window for responding and draft updating is closing**
>
> Dear Reviewer #vHVL,
>
> Thanks very much for your time and valuable comments. We understand you're busy. But as the window for responding and paper revision is closing, would you mind checking our response ([a brief summary](https://openreview.net/forum?id=tm8s3696Ox&noteId=IxCcQNinhB), and [details](https://openreview.net/forum?id=tm8s3696Ox&noteId=uohlmGQYuZ)) and confirm whether you have any further questions? We are very glad to provide answers and revisions to your further questions.
>
> Best regards and thanks,
>
> Authors of #1615

---

> ### Author Response · Authors · 2023-11-22
> **Window for responding and draft updating is closing**
>
> Dear Reviewer #vHVL,
>
> Thanks very much for your time and valuable comments. We understand you're busy. But as the window for responding and paper revision is closing, would you mind checking our response ([a brief summary](https://openreview.net/forum?id=tm8s3696Ox&noteId=IxCcQNinhB), and [details](https://openreview.net/forum?id=tm8s3696Ox&noteId=uohlmGQYuZ)) and confirm whether you have any further questions? We are very glad to provide answers and revisions to your further questions.
>
> Best regards and thanks,
>
> Authors of #1615

---

> ### Author Response · Authors · 2023-11-22
> **Window for responding and draft updating is closing**
>
> Dear Reviewer #vHVL,
>
> Thanks very much for your time and valuable comments. We understand you're busy. But as the window for responding and paper revision is closing, would you mind checking our response ([a brief summary](https://openreview.net/forum?id=tm8s3696Ox&noteId=IxCcQNinhB), and [details](https://openreview.net/forum?id=tm8s3696Ox&noteId=uohlmGQYuZ)) and confirm whether you have any further questions? We are very glad to provide answers and revisions to your further questions.
>
> Best regards and thanks,
>
> Authors of #1615

---

### Official Review · Reviewer_qWAT · 2023-10-30

**Soundness:** 3 good
**Presentation:** 3 good
**Contribution:** 3 good
**Rating:** 6
**Confidence:** 4

**Summary:**

This work focuses on one-shot federated learning, following the common practice that uses data-free distillation with local models. Co-Boosting is proposed as an algorithm in which the synthesized data and the ensemble model collaboratively enhance one another in a progressive manner. To elaborate, the algorithm produces challenging samples by training a data generator using a loss function that is re-weighted for the samples and includes an adversarial loss component.

**Strengths:**

* The idea is quite straightforward and sound.
* Nice and clean writing
* Experiments are complete with comparison to baselines and ablations, across common datasets and FL settings

**Weaknesses:**

* The related works could be improved. Specifically,
1) FL works about distillation and ensemble with or without a server dataset: there are many follow-up works after Lin et al. (2020). More works about model ensemble or aggregation would be better.
2) Since the idea is about co-learning, it will help if a discussion about centralized co-learning techniques can be added; such as
Deep co-training for semi-supervised image recognition. CVPR 2018

* The experiments are a bit lack of insights. The paper could be stronger if deeper analysis beyond accuracy can be provided. For example, what examples are considered hard? What are the training dynamics in terms of model weighting or sample re-weighting?  Any failure cases?

=== Minor issues ===
* Suggestions on fig 1: no label for the y-axis
* Experiment tables: It is better to include the upper bound such as multi-round FedAvg or centralized training
* "round" terminology: I understand the paper is about one-shot FL but sometimes the term "round" is used in the text causing me slight confusion.

**Questions:**

Recent works in multi-round FL have suggested to use of a pre-trained model for initialization [A, B]. A pre-trained model is typically publically available on both the server side and the user side and might significantly help the convergence speed given limited rounds or distillation. Could the authors provide any insights or discussions about this case?

[A] When foundation model meets federated learning: Motivations, challenges, and future directions. ArXiv 2023
[B] On the Importance and Applicability of Pre-Training for Federated Learning. ICLR 2023

---

> ### Author Response · Authors · 2023-11-17
>
> We would like to thank the reviewer for taking the time to review our work. We appreciate that you find our paper well writing and experiments are complete. According to your valuable comments, we provide detailed feedback.
>
> **Q1**: Related works issues:
> > **a)** FL works about distillation and ensemble with or without a server dataset: there are many follow-up works after Lin et al. (2020). More works about model ensemble or aggregation would be better.
> > **b)** Since the idea is about co-learning, it will help if a discussion about centralized co-learning techniques can be added; such as Deep co-training for semi-supervised image recognition. CVPR 2018
> >
> ***Ans for Q1):***
> - **a)** We appreciate the reference to the follow-up works post Lin et al. (2020). We have added the following discussions in the revised paper. We would like to note that predominant approaches in OFL involve distilling knowledge from an ensemble using either synthetic or real server data. However, existing literature focuses on either enhancing the ensemble or the data quality in isolation. Once a higher quality of data or an improved ensemble is achieved, these elements typically remain static during the server training phase. Our work diverges from this norm by proposing a method that **periodically and mutually enhances both the ensemble and data quality**. This dynamic approach represents a significant advancement in the field of OFL. We have elaborated on this in the revision, providing a more comprehensive discussion of our methodology.
>
> - **b)** Addressing the parallels and contrasts with centralized co-learning techniques, such as deep co-training for semi-supervised image recognition (CVPR 2018), offers insightful perspectives. We have added a discussion section to our revision. Centralized co-learning is generally predicated on the concept of learning from data that offers varying perspectives. In contrast, OFL deals with more pronounced divergences in data distributions among clients. Additionally, in OFL, the models provided to the server are pre-trained, eliminating the feasibility of retraining local models.
>
> **Q2**: Experiments issues:
> > The experiments are a bit lack of insights. The paper could be stronger if deeper analysis beyond accuracy can be provided. For example, what examples are considered hard? What are the training dynamics in terms of model weighting or sample re-weighting? Any failure cases?
> >
> ***Ans for Q2):***
> Thank you for your constructive comments and kind suggestions. We appreciate your attention to the depth of our analysis and agree that a more comprehensive exploration beyond accuracy could enrich the paper.
> - We have provided preliminary insights in Figure 1 and Table 7, which highlight the effectiveness of each component of our proposed method. Regarding the dynamic model weights, it is demonstrated in Figure 1d, showcasing the performance of our weighted ensemble. Additional discussions regarding model architecture, the impact of heavier models, and the integration with advanced local training techniques are detailed in Appendix B.
> - We attribute the main success factor to the mutual boosting mechanism between synthetic data generation and the reweighting ensemble strategy. Your suggestion to delve into the nature of 'hard samples', the dynamics of model weighting, and an analysis of failure cases is indeed intriguing. We plan to incorporate these in-depth studies into our revision, as they would provide a more holistic understanding of our method.
>
> **Q3**: Minor issues:
> > **a)** Suggestions on fig 1: no label for the y-axis
> > **b)** Experiment tables: It is better to include the upper bound such as multi-round FedAvg or centralized training
> > **c)** "round" terminology: I understand the paper is about one-shot FL but sometimes the term "round" is used in the text causing me slight confusion.
> >
> ***Ans for Q3):***
>
> **a)** We thank the reviewer for pointing out the missing label on the y-axis in Figure 1. This axis represents the 'Test Accuracy,' which we have now clearly labeled and elaborated upon in the figure's caption.
>
> **b)** In the context of OFL, the upper bound of each method is inherently linked to their ensemble performance. We have added discussions in Tables 2 and Appendix B.2. The superiority of our proposed method is, in part, attributable to the enhanced performance of the ensemble. We have conducted a primary comparison with multi-round FedAvg in Appendix B.8. We agree that a comparison with centralized training is also crucial and plan to incorporate this in the revised manuscript.
>
> **c)** We apologize for any confusion caused by the use of the term "round" in the text. To clarify, in our paper, we now consistently use the term "epoch" to refer to the training epochs of the server model. We would like to note that all experiments and algorithms occur after the server has collected all client models. This means there is only one communication round in our setup.

---

> > ### Author Response · Authors · 2023-11-17
> >
> > **Q4**: Discussion on pre-trained models:
> > > Recent works in multi-round FL have suggested to use of a pre-trained model for initialization [A, B]. A pre-trained model is typically publically available on both the server side and the user side and might significantly help the convergence speed given limited rounds or distillation. Could the authors provide any insights or discussions about this case?
> > >
> > ***Ans for Q4):***
> > We appreciate the reference to recent works advocating the use of pre-trained models in multi-round Federated Learning (FL). Accordingly, we add a discussion section about the outstanding works and the promising direction. Viewing OFL through the lens of today's pre-trained foundation model era, as highlighted in the mentioned works opens up new avenues of exploration. Our work positions itself as a pioneering effort in adapting OFL to the era of foundation models. The key strength of our method is its independence from alterations in local training and its architecture-agnostic nature. This flexibility is particularly relevant in the context of foundation models, which are becoming increasingly central in various domains.

---

> ### Author Response · Authors · 2023-11-20
> **Welcome for more discussions**
>
> Dear reviewer #qWAT,
>
> Thanks for your valuable time in reviewing and insightful comments. Following your comments, we have tried our best to provide responses and revise our paper. Here is a **summary of our response** for your convenience:
>
> - (1) **Related work issues**: Following your constructive comments, we have discussed more related works. We would like to note that existing literature focuses on either enhancing the ensemble or the data quality in isolation. Once a higher quality of data or an improved ensemble is achieved, these elements typically remain static during the server training phase. Our work diverges from this norm by proposing a method that **periodically and mutually enhances both the ensemble and data quality**. We provide a comprehensive comparison and summarize them in Table 8 in response. Discussions about centralized co-learning techniques are also added to our revision to enhance our work.
> - (2) **Experiments issues**: Thank you for your constructive comments and kind suggestions. We have provided preliminary insights in Figure 1 and Table 7, which highlight the effectiveness of each component of our proposed method. We attribute the main success factor to the mutual boosting mechanism between synthetic data generation and the reweighting ensemble strategy. We also plan to incorporate more in-depth studies into our revision
> - (3) **Discussion on pre-trained models**: We appreciate the reference to recent works advocating the use of pre-trained models in multi-round Federated Learning (FL). Viewing OFL through the lens of today's pre-trained foundation model era opens up new avenues of exploration. Our work positions itself as a pioneering effort in adapting OFL to the era of foundation models. The key strength of our method is its independence from alterations in local training and its architecture-agnostic nature.
>
> We humbly hope our response has addressed your concerns. If you have any additional concerns or comments that we may have missed in our responses, we would be most grateful for any further feedback from you to help us further enhance our work.
>
> Best regards
>
> Authors of #1615

---

> ### Author Response · Authors · 2023-11-20
> **Window for responsing and draft updating is closing**
>
> Dear Reviewer #qWAT,
>
> Thanks very much for your time and valuable comments. We understand you're busy. But as the window for responding and paper revision is closing, would you mind checking our response ([a brief summary](https://openreview.net/forum?id=tm8s3696Ox&noteId=jZG4pOxvTd), and [details](https://openreview.net/forum?id=tm8s3696Ox&noteId=7DFxpCioY5)) and confirm whether you have any further questions? We are very glad to provide answers and revisions to your further questions.
>
> Best regards and thanks,
>
> Authors of #1615

---

> ### Author Response · Authors · 2023-11-21
> **Window for responding and draft updating is closing**
>
> Dear Reviewer #qWAT,
>
> Thanks very much for your time and valuable comments. We understand you're busy. But as the window for responding and paper revision is closing, would you mind checking our response ([a brief summary](https://openreview.net/forum?id=tm8s3696Ox&noteId=jZG4pOxvTd), and [details](https://openreview.net/forum?id=tm8s3696Ox&noteId=7DFxpCioY5)) and confirm whether you have any further questions? We are very glad to provide answers and revisions to your further questions.
>
> Best regards and thanks,
>
> Authors of #1615

---

> ### Author Response · Authors · 2023-11-22
> **Window for responding and draft updating is closing**
>
> Dear Reviewer #qWAT,
>
> Thanks very much for your time and valuable comments. We understand you're busy. But as the window for responding and paper revision is closing, would you mind checking our response ([a brief summary](https://openreview.net/forum?id=tm8s3696Ox&noteId=jZG4pOxvTd), and [details](https://openreview.net/forum?id=tm8s3696Ox&noteId=7DFxpCioY5)) and confirm whether you have any further questions? We are very glad to provide answers and revisions to your further questions.
>
> Best regards and thanks,
>
> Authors of #1615

---

> ### Author Response · Authors · 2023-11-22
> **Window for responding and draft updating is closing**
>
> Dear Reviewer #qWAT,
>
> Thanks very much for your time and valuable comments. We understand you're busy. But as the window for responding and paper revision is closing, would you mind checking our response ([a brief summary](https://openreview.net/forum?id=tm8s3696Ox&noteId=jZG4pOxvTd), and [details](https://openreview.net/forum?id=tm8s3696Ox&noteId=7DFxpCioY5)) and confirm whether you have any further questions? We are very glad to provide answers and revisions to your further questions.
>
> Best regards and thanks,
>
> Authors of #1615

---

> ### Author Response · Authors · 2023-11-23
> **Window for responding and draft updating is closing**
>
> Dear Reviewer #qWAT,
>
> Thanks very much for your time and valuable comments. We understand you're busy. But as the window for responding and paper revision is closing, would you mind checking our response ([a brief summary](https://openreview.net/forum?id=tm8s3696Ox&noteId=jZG4pOxvTd), and [details](https://openreview.net/forum?id=tm8s3696Ox&noteId=7DFxpCioY5)) and confirm whether you have any further questions? We are very glad to provide answers and revisions to your further questions.
>
> Best regards and thanks,
>
> Authors of #1615

---

> ### Author Response · Authors · 2023-11-23
> **Window for responding and draft updating is closing**
>
> Dear Reviewer #qWAT,
>
> Thanks very much for your time and valuable comments. We understand you're busy. But as the window for responding and paper revision is closing, would you mind checking our response ([a brief summary](https://openreview.net/forum?id=tm8s3696Ox&noteId=jZG4pOxvTd), and [details](https://openreview.net/forum?id=tm8s3696Ox&noteId=7DFxpCioY5)) and confirm whether you have any further questions? We are very glad to provide answers and revisions to your further questions.
>
> Best regards and thanks,
>
> Authors of #1615

---

### Official Review · Reviewer_WMX4 · 2023-10-31

**Soundness:** 3 good
**Presentation:** 3 good
**Contribution:** 2 fair
**Rating:** 3
**Confidence:** 4

**Summary:**

This paper proposes Co-Boosting to improve the performance of one-shot federated learning. The existing one-shot federated learning algorithm requires training a synthesized-sample generator on the server using the ensemble model of client models. The core ideas of this paper are twofold: 1. Fine-tuning the weights of the ensemble model using synthesized samples to improve its ability to classify these synthesized samples. 2. When training the generator, utilizing the output of the ensemble model to calculate the difficulty of samples, assigning higher loss function weights to hard samples. These two points can be iteratively applied, mutually enhancing each other. The experiments conducted on datasets like SVHN and CIFAR10 demonstrated that Co-Boosting has a certain superiority over existing approaches.

**Strengths:**

1. This paper pays attention to the relationship between generator training and ensemble model, which is an interesting perspective.
2. The overall paper is well organized and easy to follow.

**Weaknesses:**

1. The novelty of the proposed method is relatively insufficient. The overall method is composed of several existing tricks, lacking a comprehensive core framework or a more in-depth analysis.
2. The one-shot federated learning encompasses various types of methods, including training generators [DENSE, FedDF], uploading distilled datasets [1], utilizing existing pre-trained models [2], and so on. However, this paper specifically focuses on methods related to training generators. This point should be clearly discussed in the introduction and related work sections.
3. The experiments are insufficient. The most complex model architecture discussed in this paper is a 5-layer CNN, and the most complex dataset is CIFAR100. This raises doubts about the practical value of this one-shot method.
4. In the experimental section, this paper focuses on class distribution. However, this alone may not be sufficient to demonstrate the effectiveness of the proposed approach. This is because when there are differences in class distribution, the weights of the ensemble model will naturally play a crucial role. For instance, if a model does not support category 'a', it would be necessary to reduce the weight assigned to this model. I would recommend the authors to incorporate experiments involving feature distribution differences, for instance, by utilizing the DomainNet dataset.

[1] Zhou, Yanlin, et al. "Distilled one-shot federated learning." arXiv preprint arXiv:2009.07999 (2020).

[2] Yang, Mingzhao, et al. "Exploring One-shot Semi-supervised Federated Learning with A Pre-trained Diffusion Model." arXiv preprint arXiv:2305.04063 (2023).

**Questions:**

See Weaknesses.
In Algorithm 1 Co-Boosting, should Lines 16 to 18 be placed after Line 19?

---

> ### Author Response · Authors · 2023-11-17
>
> We would like to thank the reviewer for taking the time to review our work. We appreciate that you find our paper well organized and easy to follow. According to your valuable comments, we provide detailed feedback.
>
> **Q1**: Novelty concern:
> > "The novelty of the proposed method is relatively insufficient. The overall method is composed of several existing tricks, lacking a comprehensive core framework or a more in-depth analysis."
>
> ***Ans for Q1):***
> We apologize for the misunderstanding. In response to your valuable comments, we have added the following explanations to our revision, aiming at highlighting the novelty of this work.
>
> - We agree that we leverage a straightforward approach to realize OFL, where server model is aggregated by distilling knowledge from the ensemble as discussed in [1][3][4][5][6][7].
> - We would like to note that the main contribution is the proposed **co-boosting** approach. It is **the first time** to make the ensemble and the synthetic data mutually boost each other.
> - We believe the reviewer agrees with the motivation of the proposed method that the server model's performance is intricately linked to both the quality of synthesized data and the ensemble. Aligning with this motivation, the proposed method ensures that **the ensemble and the synthetic data are mutually promoting each other**. This dynamic interaction is the key to achieving the new state-of-the-art results, which is evidenced in Fig. 1(b)-(d) and Table 7.
> - Thanks to the simplicity design, our method **eliminates the need** for modifications to the client's local training, requires no additional data or model transmission, and accommodates heterogeneous client model architectures.
>
> We appreciate your feedback and will enhance the manuscript by providing a more detailed analysis of how this integration uniquely contributes to the field. Additionally, we will draw clearer distinctions from existing methods to further underscore the novelty and practicality of our approach.
>
> **Q2**: More related works:
> > "The one-shot federated learning encompasses various types of methods, including training generators [DENSE, FedDF], uploading distilled datasets [1], utilizing existing pre-trained models [2], and so on. However, this paper specifically focuses on methods related to training generators. This point should be clearly discussed in the introduction and related work sections"
>
> ***Ans for Q2):***
> Thank you for highlighting the breadth of methods encompassed in OFL and for bringing our attention to the related works. Accordingly, we have added the following discussions to our revision.
>
> - We have highlighted the discussions about the mentioned approaches. Specifically, we discussed various approaches, including uploading distilled datasets approach [1], using public dataset [3], uploading clusters [4], and modified local training [5][6]. In response to your valuable comments, we have highlighted the relation and difference between our work and previous works.
> -  We appreciate your mention of utilizing existing pre-trained models [2], which adopts the concept of using a pre-trained diffusion model, to generate high-quality data. We have added it in the according section.
> -  In response to your constructive comments, we have highlighted the difference between our work and the mentioned outstanding works. Specifically, our method is based on the intuition that the server model's performance is intricately linked to both the quality of synthesized data and the ensemble. Existing methods generally focus on either enhancing the ensemble or improving the synthetic data. In contrast, we propose to intertwine the ensemble and the synthetic data, fostering **mutual enhancement** (Summarized in Table 8). We believe this finding could catalyze advancements in OFL methods by underscoring the importance of optimizing their interaction. Furthermore, as discussed in the appendix (pls also refer to Table 8), only a few methods can adapt to contemporary model market scenarios, including our proposed method, due to its inherent design.

---

> > ### Author Response · Authors · 2023-11-17
> >
> > **Q3**: Experiment issues:
> > > "The experiments are insufficient. The most complex model architecture discussed in this paper is a 5-layer CNN, and the most complex dataset is CIFAR100. This raises doubts about the practical value of this one-shot method."
> >
> > ***Ans for Q3):***
> > We appreciate your comments regarding the scope of our experiments. We have highlighted and added more comprehensive experiments in the revision.
> >
> > - Our experimental setup was in line with the settings of concurrent existing works in OFL [5][6][7].
> > - We agree with your point that we should evaluate our method using different models. Accordingly, we have conducted experiments in a **client-heterogeneous** setting, where client models vary in architecture, including CNN, ResNet, MobileNet, and ShuffleNet. The results are reported in **Table 3 and Appendix B.4**. Results verify that, by integrating knowledge from these diverse architectures, our method can effectively aggregate and distill this knowledge into various server model architectures including CNN, ResNet, MobileNet and ShuffleNet.
> > - In response to your constructive comments regarding **larger models**, we have conducted experiments with more complex architectures such as ResNet and VGG, as shown in **Appendix B.5**. The superior performance in these experiments demonstrates the effectiveness of our proposed method.
> > - We would like to note that our method **does not impose any specific requirements on the model structure**. Specifically, the weighted ensemble in our approach is calculated at the logit layer, which is compatible with any model architecture. This flexibility allows our method to incorporate models with diverse architectures into a single server model without architectural constraints.
> > - We added experiments using **"Tiny-ImageNet"**, with ResNet18 serving as the client architecture backbone. We explore server models based on both CNN and ResNet18 architectures. The results presented in the following table not only confirm the effectiveness of our method but also highlight its architecture-agnostic feature. More details are discussed in **Appendix B.3**.
> >
> > **Table: Experiments on Tiny-Imagenet**
> > | Server    | α    | FedAvg       | FedDF         | F-ADI         | F-DAFL        | DENSE         | Co-Boosting   |
> > |-----------|------|--------------|---------------|---------------|---------------|---------------|---------------|
> > |  CNN      | 0.05 | -            | 6.78±0.13     | 6.44±0.78     | 6.90±0.08     | 6.71±0.14     | **7.38±0.08** |
> > |  CNN      | 0.1  | -            | 9.71±0.20     | 9.57±0.71     | 9.77±0.48     | 9.09±0.43     | **10.10±0.69**|
> > |  CNN      | 0.3  | -            | 12.62±0.45    | 11.94±0.83    | 12.65±0.54    | 12.71±0.72    | **13.83±0.84**|
> > |  ResNet   | 0.05 | 0.70±0.20    | 5.65±0.30     | 5.70±0.38     | 5.68±0.50     | 5.92±0.18     | **8.21±0.12** |
> > |  ResNet   | 0.1  | 1.19±0.84    | 8.63±0.82     | 7.01±0.26     | 7.81±0.22     | 8.88±0.23     | **10.29±0.43**|
> > |  ResNet   | 0.3  | 2.09±0.52    | 11.93±0.38    | 11.58±0.84    | 12.30±0.36    | 13.05±0.36    | **14.35±0.93**|
> >
> > Thanks again for your constructive comments, and motivating more comprehensive experiments. We believe these experiments collectively demonstrate the practical applicability of our proposed method in diverse settings and with varying model architectures, reinforcing its utility in the field of OFL.

---

> > > ### Author Response · Authors · 2023-11-17
> > >
> > > **Q4**: Feature shift settings:
> > > > "In the experimental section, this paper focuses on class distribution. However, this alone may not be sufficient to demonstrate the effectiveness of the proposed approach. This is because when there are differences in class distribution, the weights of the ensemble model will naturally play a crucial role. For instance, if a model does not support category 'a', it would be necessary to reduce the weight assigned to this model. I would recommend the authors to incorporate experiments involving feature distribution differences, for instance, by utilizing the DomainNet dataset."
> > >
> > > ***Ans for Q4):***
> > > Thank you for your insightful suggestion regarding the exploration of feature distribution differences. Accordingly, we **extend our experiments to include commonly used domain generalization datasets**, specifically MNIST-M and PACS. These experiments follow the settings outlined in [8][9], which focus on federated domain generalization. In these experiments, each domain is allocated to a different client, and we employ the leave-one-domain-out testing. This approach allows us to assess our method's performance under scenarios of feature distribution shifts in the OFL setting.
> > >
> > > For these experiments, we utilized a CNN backbone for MNIST-M and ResNet18 for PACS. To further evaluate our method's versatility, we tested with both CNN and ResNet18 as server architectures. The results presented in the following table demonstrate the superior performance of our proposed method, which we attribute to the mutual enhancement principle inherent in our approach. This principle not only validates our method's model-agnostic feature but also its effectiveness in settings with feature distribution shifts. More details are added in **Appendix B.3**. The performance gains achieved by our method align with the observations in [9], which demonstrates that adjusting client weights can mitigate the challenges posed by feature shifts.
> > >
> > > **Table: Experiments on MNIST-M**
> > > | Server | Domain       | FedAvg      | FedDF        | F-ADI        | F-DAFL       | DENSE        | Co-Boosting  |
> > > |--------|--------------|-------------|--------------|--------------|--------------|--------------|--------------|
> > > | CNN    | MNIST        | 25.82±1.40  | 66.38±1.37   | 66.27±1.48   | 66.75±1.13   | 70.81±1.12   | **82.15±0.83** |
> > > | CNN    | MNIST-M      | 16.40±1.61  | 41.18±0.34   | 39.34±1.50   | 39.85±1.02   | 41.35±1.19   | **42.85±0.80** |
> > > | CNN    | SVHN         | 18.88±1.72  | 47.91±1.78   | 45.38±1.86   | 45.05±0.83   | 46.20±1.10   | **53.23±1.67** |
> > > | CNN    | SYN          | 45.97±1.96  | 78.91±0.29   | 78.70±0.57   | 79.83±0.55   | **79.90±0.30** | 78.67±0.69   |
> > > | CNN    | **AVG**          | 26.77±1.67  | 58.59±1.12   | 57.39±1.35   | 57.87±0.88   | 59.57±0.93   | **64.23±1.00** |
> > > | ResNet | MNIST        | -           | 72.68±1.53   | 70.91±1.23   | 69.15±2.06   | 73.25±1.59   | **84.39±2.31** |
> > > | ResNet | MNIST-M      | -           | 43.42±0.54   | 42.40±1.22   | 41.69±1.34   | 43.71±1.56   | **45.25±1.03** |
> > > | ResNet | SVHN         | -           | 47.91±0.88   | 48.02±0.70   | 48.46±1.48   | 47.27±1.19   | **53.75±1.13** |
> > > | ResNet | SYN          | -           | 80.48±0.51   | 79.49±1.54   | **80.55±0.88** | 80.03±0.44   | 77.93±1.22   |
> > > | ResNet | **AVG**          | -           | 61.12±0.86   | 60.21±1.17   | 59.96±1.44   | 61.07±1.20   | **65.33±1.42** |
> > >
> > > **Table: Experiments on PACS**
> > > | Server  | Domain | FedAvg      | FedDF        | F-ADI        | F-DAFL       | DENSE        | Co-Boosting  |
> > > |---------|--------|-------------|--------------|--------------|--------------|--------------|--------------|
> > > | CNN     | P      | -           | 33.80±0.45   | 32.71±1.53   | 34.25±2.12   | 35.68±1.83   | **50.77±1.78** |
> > > | CNN     | A      | -           | 22.56±0.43   | 21.17±0.52   | 21.14±0.78   | 22.41±0.39   | **23.54±1.18** |
> > > | CNN     | C      | -           | 29.65±0.30   | 29.32±0.83   | 29.61±0.86   | 29.01±0.99   | **37.59±1.41** |
> > > | CNN     | S      | -           | 30.45±0.49   | 29.85±0.26   | 28.36±0.76   | 30.17±0.34   | **30.67±2.67** |
> > > | CNN     | **AVG**    | -           | 29.12±0.42   | 28.26±0.79   | 28.34±1.13   | 29.34±0.89   | **35.64±1.66** |
> > > | ResNet  | P      | 11.32±1.93  | 37.09±0.45   | 36.50±1.04   | 37.55±1.78   | 37.19±1.89   | **51.43±1.72** |
> > > | ResNet  | A      | 18.50±2.88  | 23.76±0.85   | 22.95±0.82   | 22.92±1.89   | 24.83±1.49   | **26.76±1.23** |
> > > | ResNet  | C      | 16.60±1.28  | 29.62±0.48   | 28.53±0.89   | 29.35±1.68   | 31.78±1.30   | **36.73±1.24** |
> > > | ResNet  | S      | 19.65±2.63  | 30.20±0.42   | 29.77±0.85   | 29.67±0.76   | 32.00±1.20   | **35.35±1.74** |
> > > | ResNet  | **AVG**    | 16.52±2.18  | 30.16±0.55   | 29.31±0.90   | 29.87±1.53   | 31.45±1.47   | **37.56±1.48** |
> > >
> > > We are grateful for your recommendation and believe that these additional experiments strengthen the overall validation of our approach in the OFL landscape.

---

> > > > ### Author Response · Authors · 2023-11-17
> > > >
> > > > **Q5**: Algorithm issues:
> > > > > "In Algorithm 1 Co-Boosting, should Lines 16 to 18 be placed after Line 19?"
> > > >
> > > > ***Ans for Q5):***
> > > > We appreciate your query regarding the sequence of steps in Algorithm 1. We re-check the algorithm and confirm that the algorithm is correctly presented, and its structure is integral to the **mutual enhancement** spirit at the core of our approach. Traditional approaches typically follow a sequential process. They first focus on generating high-quality data or creating a high-quality ensemble model. Once this step is achieved, knowledge is distilled from these high-quality sources into the server model. This process is linear, where the enhancement of data or the ensemble is a prerequisite step before the server model can benefit.
> > > >
> > > > In contrast, as illustrated in Fig. 1a, our method introduces **an iterative and interconnected approach**. Instead of treating the generation of high-quality data or ensemble as isolated, preliminary steps, we integrate these processes into a continuous, dynamic cycle. Our algorithm begins by generating 'hard' samples based on the current state of the weighted ensemble and the server model. These challenging samples are then used to adjust the weights of the ensemble. Subsequently, the server model is updated by distilling knowledge from both the enriched data and the refined ensemble. This process is not a one-time enhancement but an ongoing cycle where the quality of data and the ensemble are simultaneously and iteratively improved in each epoch.
> > > >
> > > > Figure 1d effectively demonstrates that as the epochs progress, our weighted ensemble becomes increasingly refined, leading to a better-performing server model, aided by the periodic generation of harder samples. This paradigm underscores the unique contribution of our method, which innovatively fosters a dynamic, mutual enhancement of both the ensemble and the data in OFL. This dynamic interaction, we believe, is key to achieving the new state-of-the-art results demonstrated in our research.
> > > >
> > > > Therefore, the placement of Lines 16 to 18 before Line 19 in Algorithm 1 is deliberate and crucial for reflecting the iterative and interconnected nature of our method's core mechanism.
> > > >
> > > > >  ***Reference***
> > > > >
> > > > > [1] Distilled one-shot federated learning. In ArXiv preprint 2020.
> > > > >
> > > > > [2] Exploring One-shot Semi-supervised Federated Learning with A Pre-trained Diffusion Model. In ArXiv preprint 2022.
> > > > >
> > > > > [3] Practical one-shot federated learning for cross-silo setting. In IJCAI 2021.
> > > > >
> > > > > [4] Heterogeneity for the win: One-shot federated clustering. In ICML 2021.
> > > > >
> > > > > [5] Towards addressing label skews in one-shot federated learning. In ICLR 2023.
> > > > >
> > > > > [6] Data-free one-shot federated learning under very high statistical heterogeneity. In ICLR 2023.
> > > > >
> > > > > [7] Dense: Data-free one-shot federated learning. In NIPS 2022.
> > > > >
> > > > > [8] Federated Domain Generalization: A Survey. In Arxiv preprint 2023.
> > > > >
> > > > > [9] Federated Domain Generalization with Generalization Adjustment. In CVPR 2023.

---

> ### Author Response · Authors · 2023-11-20
> **Welcome for more discussions**
>
> Dear reviewer #WMX4,
>
> Thanks for your valuable time in reviewing and insightful comments. Following your comments, we have tried our best to provide responses and revise our paper. Here is a **summary of our response** for your convenience:
>
> - (1) **Novelty concern**: We would like to note that the main contribution is the proposed co-boosting approach. It is **the first time** to make the ensemble and the synthetic data **mutually boost each other**. Our method also **eliminates the need** for modifications to the client's local training, requires no additional data or model transmission, and accommodates heterogeneous client model architectures. We add these discussions into our revision to enhance our work.
> - (2) **Related work issues**: Following your constructive comments, we have discussed related works including uploading distilled datasets approach, using public datasets, uploading clusters, modifying local training, and utilizing existing pre-trained models to highlight our novelty. We provide a comprehensive comparison and summarize them in Table 8 in response.
> - (3) **More experiments**: Thanks for your valuable comments, we have provided the inherent **model structure-agnostic** feature in a client-heterogeneous setting, results in Table 3 and Appendix B.4. We also conduct experiments to evaluate our works in larger models. Results are shown in Appendix B.5. Regarding more datasets, we operate experiments using "Tiny-ImageNet". Results shown in Appendix B.3 and the following illustrate that Co-Boosting outperforms existing related works.
> - (4) **Feature shift exploration**: Thank you for your insightful suggestion regarding the exploration of feature distribution differences. We have extended our experiments to include commonly used domain generalization datasets, MNIST-M and PACS. Results in Appendix B.3 and the following demonstrate the superior performance of our proposed method, which we attribute to the mutual enhancement principle inherent in our approach.
>
> We humbly hope our response has addressed your concerns. If you have any additional concerns or comments that we may have missed in our responses, we would be most grateful for any further feedback from you to help us further enhance our work.
>
> Best regards
>
> Authors of #1615

---

> ### Author Response · Authors · 2023-11-20
> **Window for responsing and draft updating is closing**
>
> Dear Reviewer #WMX4,
>
> Thanks very much for your time and valuable comments. We understand you're busy. But as the window for responding and paper revision is closing, would you mind checking our response ([a brief summary](https://openreview.net/forum?id=tm8s3696Ox&noteId=x1SAGpc1jK), and [details](https://openreview.net/forum?id=tm8s3696Ox&noteId=5hVsJeNw6H)) and confirm whether you have any further questions? We are very glad to provide answers and revisions to your further questions.
>
> Best regards and thanks,
>
> Authors of #1615

---

> ### Author Response · Authors · 2023-11-21
> **Window for responding and draft updating is closing**
>
> Dear Reviewer #WMX4,
>
> Thanks very much for your time and valuable comments. We understand you're busy. But as the window for responding and paper revision is closing, would you mind checking our response ([a brief summary](https://openreview.net/forum?id=tm8s3696Ox&noteId=x1SAGpc1jK), and [details](https://openreview.net/forum?id=tm8s3696Ox&noteId=5hVsJeNw6H)) and confirm whether you have any further questions? We are very glad to provide answers and revisions to your further questions.
>
> Best regards and thanks,
>
> Authors of #1615

---

> ### Author Response · Authors · 2023-11-22
> **Window for responding and draft updating is closing**
>
> Dear Reviewer #WMX4,
>
> Thanks very much for your time and valuable comments. We understand you're busy. But as the window for responding and paper revision is closing, would you mind checking our response (a brief summary and details) and confirming whether you have any further questions?
>
> We are looking forward to your kind comments and questions.
>
> Best regards and thanks,
>
> Authors of #1615

---

> ### Author Response · Authors · 2023-11-22
> **Window for responding and draft updating is closing**
>
> Dear Reviewer #WMX4,
>
> Thanks very much for your time and valuable comments. We understand you're busy. But as the window for responding and paper revision is closing, would you mind checking our response (a brief summary and details) and confirming whether you have any further questions?
>
> We are looking forward to your kind comments and questions.
>
> Best regards and thanks,
>
> Authors of #1615

---

> ### Author Response · Authors · 2023-11-23
> **Window for responding and draft updating is closing**
>
> Dear Reviewer #WMX4,
>
> Thanks very much for your time and valuable comments. We understand you're busy. But as the window for responding and paper revision is closing, would you mind checking our response (a brief summary and details) and confirming whether you have any further questions?
>
> We are looking forward to your kind comments and questions.
>
> Best regards and thanks,
>
> Authors of #1615

---

> ### Author Response · Authors · 2023-11-23
> **Window for responding and draft updating is closing**
>
> Dear Reviewer #WMX4,
>
> Thanks very much for your time and valuable comments. We understand you're busy. But as the window for responding and paper revision is closing, would you mind checking our response (a brief summary and details) and confirming whether you have any further questions?
>
> We are looking forward to your kind comments and questions.
>
> Best regards and thanks,
>
> Authors of #1615

---

### Meta-Review · Area_Chair_o6ZY · 2023-12-06

**Metareview:**

The presented work introduces Co-Boosting, a novel framework designed to enhance the effectiveness of one-shot federated learning. The existing one-shot federated learning algorithm involves training a synthesized-sample generator on the server using the ensemble model of client models. Co-Boosting improves upon this by fine-tuning the weights of the ensemble model with synthesized samples, thereby enhancing its ability to classify them. Additionally, during the training of the generator, the output of the ensemble model is utilized to assess the difficulty of samples, assigning higher loss function weights to challenging samples. This iterative process of mutual enhancement between the ensemble model and synthesized data results in improved overall performance. The proposed algorithm is validated through experiments on datasets such as SVHN and CIFAR10, demonstrating its superiority over existing approaches. Co-Boosting stands out as a promising approach in one-shot federated learning, emphasizing the collaborative progression of synthesized data and the ensemble model. The algorithm produces challenging samples by incorporating a re-weighted loss function and an adversarial loss component, contributing to its effectiveness in enhancing the learning process.

**Justification For Why Not Higher Score:**

1. The authors have not addressed concerns regarding the experimental validation when it comes to slightly more complex datasets such as Tiny-Imagenet, where performances remain very low, raising doubts about the value of training-generator-based methods and the method's practical applicability

2. The proposed method shows sensitivity to class distribution, with a diminishing performance gap as class differences decrease. This should be explored further and analysed

**Justification For Why Not Lower Score:**

The extensive rebuttal has addressed most of the concerns except those mentioned in the previous section.

---

### Decision · Program_Chairs · 2024-01-16

Accept (poster)